# (Doubly) Exponential Lower Bounds for Follow the Regularized Leader in Potential Games

Ioannis Anagnostides [* 1]  Ioannis Panageas [2]  Nikolas Patris [* 2]  Tuomas Sandholm [1 3 4 5]

## Abstract

*Follow the regularized leader* (`FTRL`) is the premier algorithm for online optimization. However, despite decades of research on its convergence in constrained optimization—and potential games in particular—its behavior remained hitherto poorly understood. In this paper, we establish that `FTRL` can take exponential time to converge to a Nash equilibrium in two-player potential games for any (permutation-invariant) regularizer and potentially vanishing learning rate. By known equivalences, this translates to an exponential lower bound for certain mirror descent counterparts, most notably multiplicative weights update. On the positive side, we establish the potential property for `FTRL` and obtain an exponential upper bound $\exp\big(O_\epsilon(1/\epsilon^2)\big)$ for any no-regret dynamics executed in a lazy, alternating fashion, matching our lower bound up to factors in the exponent. Finally, in multi-player potential games, we show that fictitious play—the extreme version of `FTRL`—can take *doubly* exponential time to reach a Nash equilibrium. This constitutes an exponentially stronger lower bound for the foundational learning algorithm in games.

## 1. Introduction

*Multiplicative weights update (`MWU`)* is the quintessential online algorithm (Littlestone & Warmuth, 1994), attaining the optimal *regret* bound in many fundamental online learning problems (Fan et al., 2025). It has found applications in diverse areas ranging from approximation algorithms to boosting in machine learning (Arora et al., 2012). `MWU` is an instantiation of the seminal online learning paradigm *follow*

---
*Equal contribution [1]Carnegie Mellon University [2]University of California, Irvine [3]Strategy Robot, Inc. [4]Strategic Machine, Inc. [5]Optimized Markets, Inc.. Correspondence to: Ioannis Anagnostides <ianagnos@cs.cmu.edu>, Nikolas Patris <npatris@uci.edu>.

*Proceedings of the 43rd International Conference on Machine Learning*, Seoul, South Korea. PMLR 306, 2026. Copyright 2026 by the author(s).

*the regularized leader* (`FTRL`) (Kalai & Vempala, 2005). Viewed differently, `MWU` can also be cast as an instance of online *mirror descent* (`MD`) (Nemirovski & Yudin, 1983).

A celebrated connection inextricably links the no-regret property, attained by algorithms such as `FTRL` and `MD`, to *coarse correlated equilibria (CCEs)*, a seminal game-theoretic solution concept (Moulin & Vial, 1978; Aumann, 1974). Specifically, if each player in a multi-player game employs a no-regret algorithm to update its strategy, the *average* correlated distribution of play converges to the set of CCEs. This connection, however, provides no information about the *last-iterate* convergence of the dynamics.

Characterizing the convergence of `MWU` and other online algorithms has received extensive attention. Kleinberg et al. (2009) first established that `MWU` converges to Nash equilibria for almost all *potential* games. In this setting, players are effectively ascending a global potential function; thus, the dynamics can be viewed through the lens of constrained optimization, with Nash equilibria corresponding to first-order stationary points. Non-asymptotic rates soon emerged for some versions of `MD`, most notably gradient descent, but despite decades of research on this topic, a major question remained hitherto wide open:

> *How many iterations are needed for `FTRL` algorithms to converge in potential games?*

Viewed differently, can `MWU` or `FTRL` efficiently find first-order stationary points in constrained optimization?

Although `MWU` has been shown to asymptotically converge in potential games—at least when the set of Nash equilibria comprises isolated points, a condition that holds for almost all potential games (Kleinberg et al., 2009; Palaiopanos et al., 2017)—this result has not been extended to the general class of `FTRL`. `FTRL` presents distinct challenges that complicate the analysis compared to algorithms such as gradient descent. For example, it can exhibit *spurious fixed points*: due to its history dependence, the dynamics may stagnate even when the current strategy is highly suboptimal relative to the observed utility (Example 3.3).

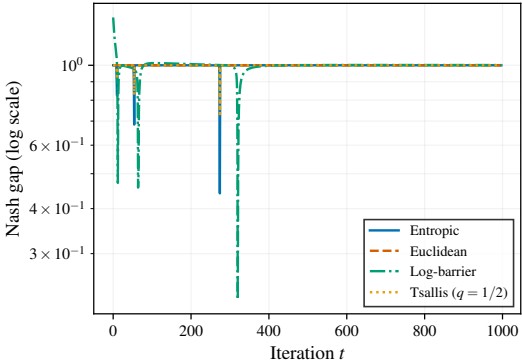 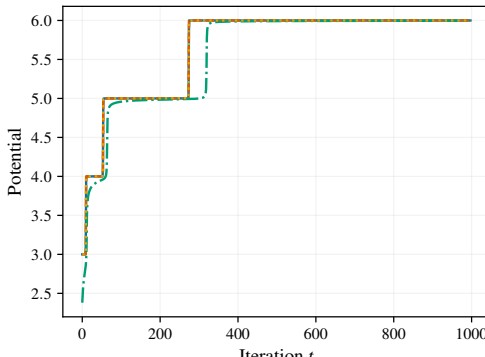

*Figure 1.* Illustration of our results: FTRL algorithms have the potential property (right), but can take exponential time to converge (left).

## 1.1. Our results

We provide new upper and lower bounds for the convergence of FTRL in potential games and constrained optimization.

Our first positive result goes beyond FTRL and covers the entire class of no-regret dynamics. We establish a non-asymptotic rate of convergence to Nash equilibria, provided the updates are alternating and *lazy*—the player performs an update only when the proposed strategy guarantees a sufficient improvement.

**Theorem 1.1.** *In any potential game, alternating $\epsilon$-lazy no-regret dynamics converge to an $\epsilon$-Nash equilibrium after at most $\exp\left(O_\epsilon(1/\epsilon^2)\right)$ iterations.*

The proof proceeds by bounding the number of updates that can occur and then deriving a recursion for the time required between successive updates, paving the way to Theorem 1.1. To put this into context, it is well-known that no-regret dynamics can fail to converge to Nash equilibria in potential games (Kleinberg et al., 2009); Theorem 1.1 offers a natural way to circumvent those impossibility results.

Turning to the special case of FTRL, we show that, for sufficiently small learning rate, the potential value is nondecreasing (Figure 1, right), and this does not require alternation nor laziness (Proposition 3.2). As a result, FTRL is bound to converge to a set in which the value of the potential is constant.

While Theorem 1.1 only establishes an exponential upper bound in $1/\epsilon^2$, our next result justifies this by establishing a fundamental barrier for FTRL dynamics, which manifests itself with or without alternation or laziness.

**Theorem 1.2.** *In two-player $m \times m$ potential games, FTRL can take $2^{\Omega(m \log m)}$ iterations to converge to an $\epsilon$-Nash equilibrium for $\epsilon = 1/\text{poly}(m)$. This holds for any permutation-invariant regularizer and learning rate $\eta^{(t)} = 1/t^\alpha$ for $\alpha \in [0, 1)$.*

Crucially, this theorem applies to *any* member of FTRL

with a permutation invariant regularizer, which encompasses most common incarnations of FTRL (Figure 1, left). By virtue of known equivalences, Theorem 1.2 thereby circumscribes specific MD algorithms as well, most notably MWU. Furthermore, Theorem 1.2 holds even under a vanishing learning rate $\eta^{(t)} = 1/t^\alpha$ for $\alpha \in [0, 1)$. The analytical challenge in this regime stems from the propensity of FTRL to mix actions, especially when the learning rate is close to zero. Another notable aspect of our lower bound construction is that it tightly mirrors the mechanism driving our upper bound.

Finally, we turn our attention to $n$-player potential games. We analyze *fictitious play (FP)* (Robinson, 1951; Brown, 1951), the foundational learning algorithm in games, which can be viewed as the extreme version of FTRL when $\eta \to \infty$. We show that FP can take *doubly* exponentially many iterations to converge to a Nash equilibrium.

**Theorem 1.3.** *There is an $n$-player binary-action potential game in which FP takes $2^{\Omega(2^n)}$ iterations to reach an $\epsilon$-Nash equilibrium for $\epsilon = 1/2^n$.*

This constitutes an exponentially stronger lower bound relative to prior work (Panageas et al., 2023; Brandt et al., 2013). From a technical standpoint, our multi-player construction distills the essential property behind the two-player lower bound by making a connection to a classic graph-theoretic problem connected to coding theory (Section 5).

## 1.2. Related work

**Learning in potential games**   There is a rich literature on the convergence of learning dynamics in potential games. Unlike general games, potential games always admit a *pure* Nash equilibrium, and better-response dynamics converge to one (Rosenthal, 1973; Monderer & Shapley, 1996). Fictitious play is also known to converge in this setting, though it was recently shown that it may require exponential time to do so (Panageas et al., 2023). Our lower bound in two-player games adapts their construction; unlike fictitious play, FTRL

has, by definition, a propensity to mix, which significantly complicates the analysis. The construction of Panageas et al. (2023) was also recently used by Anagnostides et al. (2025) to show that *regret matching* (Hart & Mas-Colell, 2000) can also require exponential time to converge.

Beyond these classic dynamics, significant effort has been devoted to analyzing other learning algorithms in potential games (Héliou et al., 2017; Palaiopanos et al., 2017; Anagnostides et al., 2022; Hart & Mas-Colell, 2003; Marden et al., 2007; Candogan et al., 2013; Maheshwari et al., 2025). From a complexity standpoint, computing Nash equilibria in potential games is likely intractable when the precision is exponentially small (Babichenko & Rubinstein, 2021; 2020), but amenable to algorithms such as gradient descent when the precision is inverse polynomial. Our results reveal that FTRL algorithms are poor first-order optimizers even in the inverse polynomial regime.

From a broader vantage point, the convergence of FTRL in games is a topic of active research (Lotidis et al., 2025a;b; Kamp et al., 2025).

**Fictitious play** Following the pioneering work of Robinson (1951), the convergence of FP has received extensive attention in two-player zero-sum games. Daskalakis & Pan (2014) proved an exponential lower bound, albeit under adversarial tie breaking. Characterizing its convergence beyond adversarial tie breaks remains open, although Wang (2025) recently made progress. Our construction holds regardless of how ties are broken.

**Forgetfulness and convergence speed** Our finding that FTRL is exponentially slower than MD in potential games mirrors a recent result by Cai et al. (2024), albeit in a fundamentally different problem and class of algorithms. Specifically, their result pertains to (two-player) zero-sum games and *optimistic* algorithms.

## 2. Preliminaries

Our paper revolves around the FTRL algorithm, which is reviewed in Section 2.1. Section 2.2 provides some basic background on potential games.

### 2.1. Online learning and FTRL

We examine the convergence of *follow the regularized leader (FTRL)*. This is a classic online algorithm introduced by Kalai & Vempala (2005), which became a mainstay in online optimization (Shalev-Shwartz, 2012). Taking a step back, in the usual online learning framework, a learner interacts with an environment over a sequence of $T$ rounds. In each round $t \in [T]$, the learner selects a strategy $\boldsymbol{x}^{(t)} \in \mathcal{X}$, where $\mathcal{X}$ is a convex and compact set such as

the probability simplex. The environment then specifies a utility vector $\boldsymbol{u}^{(t)}$, so that the utility of the learner in round $t$ reads $\langle \boldsymbol{x}^{(t)}, \boldsymbol{u}^{(t)} \rangle$. FTRL is a specific update rule that maps the history of observed utilities to the next strategy of the learner. Specifically, upon observing a sequence of utilities $(\boldsymbol{u}^{(\tau)})_{\tau=1}^{t}$, it computes

$$\boldsymbol{x}^{(t+1)} := \arg\max_{\boldsymbol{x} \in \mathcal{X}} \left\{ \left\langle \boldsymbol{x}, \sum_{\tau=1}^{t} \boldsymbol{u}^{(\tau)} \right\rangle - \frac{1}{\eta^{(t)}} \mathcal{R}(\boldsymbol{x}) \right\}. \quad (1)$$

Here, $\mathcal{R}$ is a 1-strongly convex and continuously differentiable[1] regularizer with respect to some norm $\|\cdot\|$: $\mathcal{R}(\boldsymbol{x}) \geq \mathcal{R}(\boldsymbol{x}') + \langle \nabla \mathcal{R}(\boldsymbol{x}'), \boldsymbol{x} - \boldsymbol{x}' \rangle + \frac{1}{2} \|\boldsymbol{x} - \boldsymbol{x}'\|^2$ for any $\boldsymbol{x}, \boldsymbol{x}'$; and $\eta^{(t)} > 0$ is the (nonincreasing) *learning rate* sequence. When $\eta^{(t)} = +\infty$, FTRL approaches *fictitious play* (Robinson, 1951)—also known as *follow the leader* in the online learning literature—formally defined in the sequel. We will denote by $R$ the *range* of the regularizer $\mathcal{R}$, that is, $|\mathcal{R}(\boldsymbol{x}) - \mathcal{R}(\boldsymbol{x}')| \leq R$ for any $\boldsymbol{x}, \boldsymbol{x}' \in \mathcal{X}$. Also, $\|\cdot\|_*$ denotes the dual norm of $\|\cdot\|$; that is, $\|\boldsymbol{u}\|_* = \sup_{\|\boldsymbol{x}\| \leq 1} \langle \boldsymbol{x}, \boldsymbol{u} \rangle$.

A classic result due to Kalai & Vempala (2005) shows that, unlike fictitious play, FTRL has the *no-regret* property even against an adversarially produced sequence of utilities. Regret is the most common way of measuring performance in an online learning environment. It is defined as

$$\mathsf{Reg}^{(T)} = \max_{\boldsymbol{x}' \in \mathcal{X}} \sum_{t=1}^{T} \langle \boldsymbol{x}' - \boldsymbol{x}^{(t)}, \boldsymbol{u}^{(t)} \rangle. \quad (2)$$

We are now ready to formally state the regret guarantee of FTRL, as defined in (1).

**Proposition 2.1** (Kalai & Vempala, 2005; Shalev-Shwartz, 2012). *For any sequence of utilities* $(\boldsymbol{u}^{(t)})_{t=1}^{T}$, *the regret of* FTRL *can be bounded as*

$$\mathsf{Reg}^{(T)} \leq \frac{R}{\eta^{(T)}} + \sum_{t=1}^{T} \eta^{(t)} \|\boldsymbol{u}^{(t)}\|_*^2$$

*for any nonincreasing learning rate sequence* $\eta^{(t)}$. *In particular, if* $\|\boldsymbol{u}^{(t)}\|_* \leq B$ *for all* $t$ *and* $\eta^{(t)} = C(m,B)/t^\alpha$ *for some* $\alpha \in (0,1)$,

$$\mathsf{Reg}^{(T)} \leq \frac{RT^\alpha}{C(m,B)} + B^2 C(m,B) \left( 1 + \frac{T^{1-\alpha}}{1-\alpha} \right).$$

Above, we used the fact that $\sum_{t=1}^{T} 1/t^\alpha \leq 1 + T^{1-\alpha}/(1-\alpha)$. The key takeaway of Proposition 2.1 is that $\mathsf{Reg}^{(T)} = o(T)$ for any $\alpha \in (0,1)$. We proceed to review some common instantiations of FTRL, most notably MWU.

*Example* 2.2 (MWU). A canonical member of FTRL is MWU. It is the instantiation of FTRL on the probability simplex

---

[1] Differentiability is assumed over some open set $\tilde{\mathcal{X}} \supseteq \mathcal{X}$.

$\mathcal{X} = \Delta(\mathcal{A})$ when $\mathcal{R}$ is the (negative) entropy regularizer, $\mathcal{R}(\boldsymbol{x}) = \sum_{a \in \mathcal{A}} \boldsymbol{x}[a] \log \boldsymbol{x}[a]$, which can be shown to be 1-strongly convex with respect to $\|\cdot\|_1$. In that case, FTRL admits the closed-form solution

$$\boldsymbol{x}^{(t+1)}[a] = \frac{e^{\eta^{(t)} \sum_{\tau=1}^{t} \boldsymbol{u}^{(\tau)}[a]}}{\sum_{a' \in \mathcal{A}} e^{\eta^{(t)} \sum_{\tau=1}^{t} \boldsymbol{u}^{(\tau)}[a']}} \quad \forall a \in \mathcal{A}. \quad \text{(MWU)}$$

When the learning rate is a constant, this can be equivalently written as

$$\boldsymbol{x}^{(t+1)}[a] = \boldsymbol{x}^{(t)}[a] \frac{e^{\eta \boldsymbol{u}^{(t)}[a]}}{\sum_{a' \in \mathcal{A}} \boldsymbol{x}^{(t)}[a'] e^{\eta \boldsymbol{u}^{(t)}[a']}} \quad \forall a \in \mathcal{A}.$$

Since the range of the entropy regularizer is $\log |\mathcal{A}|$, Proposition 2.1 implies that the regret of MWU is bounded by $\sqrt{T \log |\mathcal{A}|}$ when $\|\boldsymbol{u}^{(t)}\|_\infty \le 1$ for all $t$.

*Example* 2.3 (Euclidean regularization). Another canonical member of the FTRL family arises when $\mathcal{R} : \boldsymbol{x} \mapsto \frac{1}{2}\|\boldsymbol{x}\|_2^2$. In that case, the update rule can be cast as

$$\boldsymbol{x}^{(t+1)} = \Pi_{\mathcal{X}}\left(\eta \sum_{\tau=1}^{t} \boldsymbol{u}^{(\tau)}\right),$$

where $\Pi_{\mathcal{X}}(\cdot)$ denotes the Euclidean projection onto $\mathcal{X}$.

Two other notable regularizers are i) the *logarithmic regularizer* $\mathcal{R} : \boldsymbol{x} \mapsto -\sum_{a \in \mathcal{A}} \log \boldsymbol{x}[a]$, and ii) *Tsallis entropy* $\mathcal{R} : \boldsymbol{x} \mapsto -\frac{1}{q(1-q)} \sum_{a \in \mathcal{A}} (\boldsymbol{x}[a])^q$ for $q \in (0,1)$, both of which have found many applications in, among others, learning with bandit feedback (Zimmert & Seldin, 2021; Abernethy et al., 2008; Wei & Luo, 2018). All of these regularizers are permutation invariant (Definition B.1).

A regularizer $\mathcal{R}$ is called a *Legendre regularizer* if, in addition to the previous properties, for any sequence of points $\boldsymbol{x}^{(1)}, \boldsymbol{x}^{(2)}, \dots$ converging to a boundary point of $\mathcal{X}$, $\nabla \mathcal{R}(\boldsymbol{x}^{(T)}) \to \infty$ as $T \to \infty$ (Cesa-Bianchi & Lugosi, 2006). A well-known fact is that FTRL and MD are equivalent under any Legendre regularizer; for example, we refer to McMahan (2011; 2017). This means that all our results concerning FTRL—both upper and lower bounds—immediately translate to certain MD variants as well.

**Fictitious play** Fictitious play (Robinson, 1951; Brown, 1951), also known as *follow the leader* in the online learning literature, can be viewed as the extreme case of FTRL with $\eta \to \infty$. Specifically,

$$\boldsymbol{x}^{(t+1)} := \arg\max_{\boldsymbol{x} \in \mathcal{X}} \left\{ \left\langle \boldsymbol{x}, \sum_{\tau=1}^{t} \boldsymbol{u}^{(\tau)} \right\rangle \right\}.$$

When $\mathcal{X} = \Delta(\mathcal{A})$, it is assumed that fictitious play selects a pure strategy.

## 2.2. Games and solution concepts

In an $n$-player normal-form game, each player $i \in [n]$ selects as strategy a probability distribution $\boldsymbol{x}_i \in \Delta(\mathcal{A}_i) =: \mathcal{X}_i$ from a finite set of actions $\mathcal{A}_i$. We denote by $u_i(\boldsymbol{x})$ the expected utility of player $i \in [n]$ under the joint strategy $(\boldsymbol{x}_1, \dots, \boldsymbol{x}_n)$. We also use the notation $\boldsymbol{x}_{-i} = (\boldsymbol{x}_1, \dots, \boldsymbol{x}_{i-1}, \boldsymbol{x}_{i+1}, \dots, \boldsymbol{x}_n)$. The most standard solution concept in games is the *Nash equilibrium*, recalled below.

**Definition 2.4** (Nash, 1950)**.** A joint strategy $(\boldsymbol{x}_1, \dots, \boldsymbol{x}_n)$ is an $\epsilon$-*Nash equilibrium* if for any player $i \in [n]$ and deviation $\boldsymbol{x}_i' \in \mathcal{X}_i$,

$$u_i(\boldsymbol{x}_i', \boldsymbol{x}_{-i}) \le u_i(\boldsymbol{x}_i, \boldsymbol{x}_{-i}) + \epsilon.$$

As we alluded to in our introduction, there is an intimate connection between the no-regret property and a game-theoretic solution concept known as *coarse correlated equilibrium (CCE)* (Aumann, 1974; Moulin & Vial, 1978). CCEs relax Definition 2.4 by allowing for a *correlated* distribution.

**Definition 2.5.** A distribution $\mu \in \Delta(\mathcal{A}_1 \times \cdots \times \mathcal{A}_n)$ is an $\epsilon$-CCE if for any player $i \in [n]$ and deviation $\boldsymbol{x}_i' \in \mathcal{X}_i$,

$$\mathbb{E}_{\boldsymbol{a} \sim \mu}[u_i(\boldsymbol{x}_i', \boldsymbol{a}_{-i})] \le \mathbb{E}_{\boldsymbol{a} \sim \mu}[u_i(\boldsymbol{a})] + \epsilon.$$

By virtue of Proposition 2.1, it follows that FTRL dynamics converge to the set of CCEs. Specifically, if the cumulative regret of each player is bounded as $O_T(\sqrt{T})$, convergence to the set of CCEs occurs at a rate of $O_\epsilon(1/\epsilon^2)$.

**Potential games** A game is called a *potential game* if there exists a function $\Phi : \mathcal{X} \to \mathbb{R}$ such that for any player $i, \boldsymbol{x}_{-i}$, and $\boldsymbol{x}_i, \boldsymbol{x}_i' \in \Delta(\mathcal{A}_i)$,

$$u_i(\boldsymbol{x}_i, \boldsymbol{x}_{-i}) - u_i(\boldsymbol{x}_i', \boldsymbol{x}_{-i}) = \Phi(\boldsymbol{x}_i, \boldsymbol{x}_{-i}) - \Phi(\boldsymbol{x}_i', \boldsymbol{x}_{-i}).$$

This implies that the utility gradient of each player can be expressed as $\boldsymbol{u}_i(\boldsymbol{x}) = \nabla_{\boldsymbol{x}_i} \Phi(\boldsymbol{x})$. In a normal-form potential game, the potential function is multilinear, so it is smooth, but not necessarily concave. Our upper bounds apply more broadly beyond normal-form games to constrained optimization whenever the potential is $L$-*smooth*: $\|\nabla \Phi(\boldsymbol{x}) - \nabla \Phi(\boldsymbol{x}')\|_* \le L\|\boldsymbol{x} - \boldsymbol{x}'\|$ for any $\boldsymbol{x}, \boldsymbol{x}'$, where $\|\cdot\|_*$ is the dual norm. For convenience, we define $L$-smoothness with respect to the global norm $\|\boldsymbol{x}\| := \sqrt{\sum_{i=1}^{n} \|\boldsymbol{x}_i\|_{(i)}^2}$, where $\|\cdot\|_{(i)}$ is the norm induced by the regularizer $\mathcal{R}_i$ employed by player $i$. To ease the notation, we will assume that all players employ the same regularizer, although our results apply more broadly.

**Further notation** We denote by $D_i$ the diameter of $\mathcal{X}_i$ with respect to some norm $\|\cdot\|$; the choice of norm will be clear from the context. We use $B$ for an upper bound on the norm of utilities in terms of the dual norm; that is, $\|\boldsymbol{u}_i(\boldsymbol{x})\|_* \le B$ for any $i \in [n]$ and joint strategy $\boldsymbol{x}$.

# 3. Exponential upper bound and the potential property for FTRL

We begin by establishing exponential upper bounds for FTRL in potential games. Our first goal is to show that when multiple players perform an FTRL update simultaneously, the value of the potential function cannot decrease. The following lemma takes the perspective of a single player, and shows that FTRL always yields an improvement relative to its current utility.

**Lemma 3.1.** *If the sequence* $(\boldsymbol{x}_i^{(t)})_{t \geq 1}$ *is updated using* FTRL *under a 1-strongly convex regularizer* $\mathcal{R}$ *with respect to some norm* $\| \cdot \|$,

$$\langle \boldsymbol{x}_i^{(t+1)}, \boldsymbol{u}_i^{(t)} \rangle - \langle \boldsymbol{x}_i^{(t)}, \boldsymbol{u}_i^{(t)} \rangle \geq \frac{1}{\eta} \| \boldsymbol{x}_i^{(t+1)} - \boldsymbol{x}_i^{(t)} \|^2.$$

In particular, the improvement is proportional to the squared movement $\| \boldsymbol{x}_i^{(t+1)} - \boldsymbol{x}_i^{(t)} \|$. The proof of Lemma 3.1 relies on the first-order optimality conditions of the FTRL update. Together with all the other proofs of this section, it is deferred to Section A.

Lemma 3.1 already implies that FTRL can never decrease the value of the potential function under *alternating updates*, which means that players perform the update one by one; in fact, this holds no matter the choice of the learning rate.

We will now argue about the simultaneous case. Since $\Phi$ is $L$-smooth, we can lower bound the difference $\Phi(\boldsymbol{x}^{(t+1)}) - \Phi(\boldsymbol{x}^{(t)})$ by

$$\langle \nabla \Phi(\boldsymbol{x}^{(t)}), \boldsymbol{x}^{(t+1)} - \boldsymbol{x}^{(t)} \rangle - \frac{L}{2} \| \boldsymbol{x}^{(t+1)} - \boldsymbol{x}^{(t)} \|^2.$$

By Lemma 3.1, the first term can in turn be lower bounded by

$$\sum_{i=1}^{n} \langle \boldsymbol{x}_i^{(t+1)} - \boldsymbol{x}_i^{(t)}, \boldsymbol{u}_i^{(t)} \rangle \geq \frac{1}{\eta} \sum_{i=1}^{n} \| \boldsymbol{x}_i^{(t+1)} - \boldsymbol{x}_i^{(t)} \|^2.$$

As a result, when $\eta \leq 1/L$, we have shown that

$$\Phi(\boldsymbol{x}^{(t+1)}) - \Phi(\boldsymbol{x}^{(t)}) \geq \frac{1}{2\eta} \| \boldsymbol{x}^{(t+1)} - \boldsymbol{x}^{(t)} \|^2.$$

A telescopic summation yields the following implication.

**Proposition 3.2.** *Suppose that each player in a potential game employs* FTRL *with learning rate* $\eta \leq 1/L$, *where* $L$ *is the smoothness parameter of the potential. For any* $t \geq 1$,

$$\Phi(\boldsymbol{x}^{(t+1)}) - \Phi(\boldsymbol{x}^{(t)}) \geq \frac{1}{2\eta} \| \boldsymbol{x}^{(t+1)} - \boldsymbol{x}^{(t)} \|^2. \quad (3)$$

*In particular,*

$$\sum_{t=1}^{T} \sum_{i=1}^{n} \| \boldsymbol{x}_i^{(t+1)} - \boldsymbol{x}_i^{(t)} \|^2 \leq 2\eta \Phi_{\mathsf{range}}.$$

The second implication—the fact that FTRL has a bounded second-order path length—follows from (3) from a telescopic summation, noting that $\Phi_{\mathsf{range}}$ denotes the range of the potential function, which is bounded.

An interesting implication of Proposition 3.2 is that simultaneous FTRL asymptotically converges to a set in which the value of the potential is the same (Proposition A.1). Proposition 3.2 also implies that $\lim_{t \to \infty} \| \boldsymbol{x}^{(t+1)} - \boldsymbol{x}^{(t)} \| \to 0$. Specifically, for any $\epsilon > 0$, $O_\epsilon(1/\epsilon^2)$ iterations suffice so that $\| \boldsymbol{x}^{(t+1)} - \boldsymbol{x}^{(t)} \| \leq \epsilon$. For MD with a smooth regularizer, this in fact implies that $\boldsymbol{x}^{(t+1)}$ is an $O_\epsilon(\epsilon)$-Nash equilibrium (Anagnostides et al., 2022); however, this is not the case for FTRL, no matter the choice of the regularizer. In other words, FTRL can have *spurious fixed points*.

*Example* 3.3 (Spurious fixed points of FTRL). Consider a sequence of utilities $\boldsymbol{u}_i^{(\tau)} = (1, 0)$ for all $\tau \in [t]$. Under this sequence, FTRL is producing a strategy that plays the first action deterministically, up to an error proportional to $1/t$. Now, if $\boldsymbol{u}_i^{(t+1)} = (0, 1)$, it holds that $\| \boldsymbol{x}_i^{(t+1)} - \boldsymbol{x}_i^{(t)} \| \leq O_t(1/t)$, and consequently $\boldsymbol{x}_i^{(t+1)}$ is highly suboptimal relative to $\boldsymbol{u}_i^{(t+1)}$.

This inertia of FTRL is indeed what drives our lower bounds presented later in Section 4.

## 3.1. Non-asymptotic convergence of no-regret dynamics

Despite this pathological behavior, we will now establish an upper bound for FTRL. In fact, our result applies to any no-regret dynamics, provided that the updates are alternating and $\epsilon$-lazy. To put this into context, we highlight that no-regret dynamics can fail to converge in potential games (Kleinberg et al., 2009), so our positive results offer a natural way of bypassing that impossibility.

More precisely, let $\mathfrak{R}(\boldsymbol{u}_i^{(1)}, \ldots, \boldsymbol{u}_i^{(t)}) \in \mathcal{X}$ denote the output of the regret minimization algorithm upon observing the sequence $(\boldsymbol{u}_i^{(\tau)})_{\tau=1}^t$, which we assume guarantees $\mathsf{Reg}_i^{(t)} \leq O_t(t^{1-\alpha})$ for some $\alpha \in (0, 1]$. If $\tilde{\boldsymbol{x}}_i^{(t+1)} = \mathfrak{R}(\boldsymbol{u}_i^{(1)}, \ldots, \boldsymbol{u}_i^{(t)})$ at each time $t$, the strategy $\boldsymbol{x}_i^{(t+1)}$ in the $\epsilon$-*lazy* version is defined as

$$\boldsymbol{x}_i^{(t+1)} = \begin{cases} \tilde{\boldsymbol{x}}_i^{(t+1)} & \text{if } \langle \tilde{\boldsymbol{x}}_i^{(t+1)} - \boldsymbol{x}_i^{(t)}, \boldsymbol{u}_i^{(t)} \rangle \geq \epsilon, \\ \boldsymbol{x}_i^{(t)} & \text{otherwise.} \end{cases} \quad (4)$$

That is, the update occurs only if it delivers at least an $\epsilon$ improvement relative to the current utility. In this context, we prove the following result.

**Theorem 3.4.** *Alternating* $\epsilon$-*lazy no-regret dynamics converge to a* $2\epsilon$-*Nash equilibrium after at most* $\exp(O_\epsilon(1/\epsilon^2))$ *rounds.*

Our analysis proceeds by bounding the total number of updates that can occur (Lemma A.2) and the duration of the

intervals between consecutive updates (Lemma A.3). Specifically, we derive the recursion $T_{k+1} \leq T_k \left(1 + O_\epsilon \left(\frac{1}{\epsilon}\right)\right)$. What is intriguing is that the mechanism driving our upper bound mirrors the construction of our lower bound.

# 4. Exponential lower bound for FTRL

In this section, we show that FTRL can take exponentially many rounds to reach an approximate Nash equilibrium even in a two-player identical-interest games. Our lower bound applies to either alternating or simultaneous updates; in what follows, we consider simultaneous updates for concreteness. The construction is also robust under the lazy update rule in (4), establishing the tightness of Theorem 3.4 up to factors in the exponent.

In what follows, we provide a high-level overview of the main steps in our argument; the formal proofs are deferred to Section C. The class of games we study is based on the construction of Panageas et al. (2023), originally introduced to analyze fictitious play. We employ a variant of their class of games, formally defined in Section C.2. For an odd dimension $m$, we define a payoff matrix $\mathbf{A}$ whose maximum entry equals $2m - 1$, and we take the action sets to be $\mathcal{A}_1 = \mathcal{A}_2 = [m]$. For each payoff value $k \in \{3, \dots, 2m - 1\}$, we denote by $a_1(k), a_2(k) \in [m]$ the unique row and column indices, respectively, such that $\mathbf{A}[a_1(k), a_2(k)] = k$. An illustration of $\mathbf{A}$ is given in Figure 2b. The role of the extra row and column compared to $\mathbf{B}$ concerns the initialization; as we shall see, FTRL initialized uniformly at random on $\mathbf{A}$ will end up at the beginning of the spiral in the $\mathbf{B}$ submatrix.[2]

To examine the behavior of FTRL in this class of games, it is essential to keep track of the *cumulative utility gap* of each action, defined as

$$\mathrm{Gap}_i^{(t)}[a_i] := \max_{a_i' \in \mathcal{A}_i} \sum_{\tau=1}^{t} \boldsymbol{u}_i^{(\tau)}[a_i'] - \sum_{\tau=1}^{t} \boldsymbol{u}_i^{(\tau)}[a_i] \geq 0$$

for each player $i \in \{1, 2\}$. An important observation is that for FTRL algorithms, an action with large gap will always be played with small probability, as we establish in the lemma below.

**Lemma 4.1.** *Suppose that $(\boldsymbol{x}_i^{(t)})_{t \geq 1}$ is updated using FTRL. If $\mathrm{Gap}_i^{(t)}[a_i] \geq \gamma > 0$ for some action $a_i \in \mathcal{A}_i$, then*

$$\boldsymbol{x}_i^{(t+1)}[a_i] \leq \frac{R}{\eta^{(t)} \gamma}, \tag{5}$$

*where $R$ denotes the range of the regularizer.*

An important point about Lemma 4.1 is that the bound relating the gap and the probability of playing the corresponding

---

[2]For simplicity in the exposition, we allow the payoffs to be larger than 1. As we explain in Remark C.5, it is straightforward to adjust our construction even when all payoffs are in $[-1, 1]$.

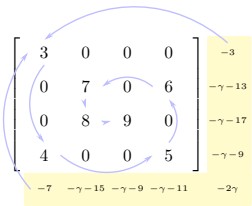

$$\begin{bmatrix} 3 & 0 & 0 & 0 \\ 0 & 7 & 0 & 6 \\ 0 & 8 & 9 & 0 \\ 4 & 0 & 0 & 5 \end{bmatrix}$$

*(a)* Original matrix $\mathbf{B}$.   *(b)* New matrix $\mathbf{A}$.

action in (5) becomes *weaker* as the learning rate gets closer to 0, which happens at a rate of $1/t^\alpha$. In the beginning, $\eta^{(t)}$ is relatively large, so the regularizer has a limited impact. As the learning rate decreases, the propensity of FTRL to mix over actions intensifies, introducing a considerable obstacle in the lower bound construction; it is harder to control the trajectory of FTRL under significant mixing. Our key observation is that the gap for the relevant actions will grow *linearly* in $t$ (Lemma 4.4), thereby subsuming the vanishing effect of the learning rate in (5).

**Periods**  The key invariance we establish is that the trajectory of FTRL can be divided into $k$ *periods*, so that in the $k$th period FTRL places high probability on the profile $(a_1(k), a_2(k))$. Because FTRL has a tendency to mix, especially under Legendre regularizers, our analysis needs to account for the residue probability. To do so, we fix the threshold $\delta := \frac{1}{4m}$ and define the periods as follows.

**Definition 4.2** (Transition period). *For each integer $k \in \{3, 4, \dots, 2m - 1\}$, the transition period $k$ is the set of rounds $t$ for which the following conditions hold:*

(i) *If $k \geq 4$ is even, then for all $t \in [\underline{t_k}, \overline{t_k}]$, $\boldsymbol{x}_2^{(t)}[a_2(k)] \geq 1 - \delta$ and $\boldsymbol{x}_1^{(t)}[a_1(k-1)] + \boldsymbol{x}_1^{(t)}[a_1(k)] \geq 1 - \delta$.*

(ii) *If $k \geq 5$ is odd, then for all $t \in [\underline{t_k}, \overline{t_k}]$, $\boldsymbol{x}_1^{(t)}[a_1(k)] \geq 1 - \delta$ and $\boldsymbol{x}_2^{(t)}[a_2(k-1)] + \boldsymbol{x}_2^{(t)}[a_2(k)] \geq 1 - \delta$.*

For $k \geq 3$, we let $\underline{t_k}$ and $\overline{t_k}$ denote the first and last rounds of period $k$, respectively. $T_k := \overline{t_k} - \underline{t_k} + 1$ denotes its length, so that period $k$ corresponds to the interval $t \in [\underline{t_k}, \overline{t_k}]$.

A key property that we establish is that FTRL transitions from each period $k$ to its successor period $k + 1$.

**Property 4.3** (Period consistency). *Suppose both players employ FTRL. If a round $t$ belongs to period $k$ for some $3 \leq k \leq 2m - 1$, then the subsequent round $t + 1$ belongs either to the same period $k$ or to the next period $k + 1$.*

In other words, there are no shortcuts: FTRL needs to traverse the entire path to reach the maximum payoff.

To argue about Property 4.3, let us say that $k$ is even. We need to analyze the transition of the row player from $a_1(k-$

1) to $a_1(k)$. When the row player starts playing $a_1(k)$ with higher and higher probability, it triggers a change in the utility observed by the column player. What we need to show is that the transition from $a_1(k-1)$ to $a_1(k)$ will be faster than the time it takes for the column player to react.

This is indeed the case for the following reason. The transition from $a_1(k-1)$ to $a_1(k)$ takes time proportional to $1/\eta^{(t)}$, which is of the order of $t^\alpha$. On the other hand, the time it takes for the column player to react is $\Omega(t)$. This is a high-level simplification; the precise argument is given in Section C.6.

**Growth of the cumulative utility gaps**  Having established this key invariance, we show that the gap of actions corresponding to future periods grows considerably.

**Lemma 4.4.** *Let $k \geq 6$ be even. Then*

$$\text{Gap}_1^{(\overline{t_{k-2}})}[a_1(k)] \geq \sum_{\ell=4}^{k-2} \Big[ (1-\delta)(\ell-1) - \delta(2m-1) \Big] T_\ell.$$

*Similarly, if $k \geq 5$ is odd, then*

$$\text{Gap}_2^{(\overline{t_{k-2}})}[a_2(k)] \geq \sum_{\ell=4}^{k-2} \Big[ (1-\delta)(\ell-1) - \delta(2m-1) \Big] T_\ell.$$

When $\delta$ is small enough, we use Lemma 4.4 to derive the following recursion.

**Lemma 4.5** (Recurrence relation of $T_k + T_{k-1}$). *For a small enough $\delta > 0$,*

$$T_k + T_{k-1} \geq \frac{1}{2} \sum_{\ell=4}^{k-2} (\ell - 2) T_\ell$$

*for all $k \geq 6$.*

The only way for FTRL to have a small Nash equilibrium gap under the invariance established in Property 4.3 is for it to reach the last period (Lemma C.13). However, on account of Lemma 4.5, this takes $2^{\Omega(m \log m)}$ rounds, establishing our lower bound, as formalized in Theorem C.14.

**Basis of the induction**  The validity of our previous argument rests on $t$ being large enough, so that the linear gap $\Omega(t)$ subsumes factors that grow as $t^\alpha$. To complete the proof, we establish the basis of the induction. Specifically, we construct a suitable gadget that forces FTRL to i) immediately transition at the beginning of the spiral and ii) spend a considerable amount of time in the initial period. This can be achieved by appending an additional row and column to the matrix with sufficiently negative entries (Figure 2b). When FTRL is initialized uniformly at random, which is the optimal choice for permutation-invariant

regularizers (Lemma B.2), all actions except the first will experience large regret. This means that escaping from the initial period will require many rounds so as to overcome the negative regret. Setting the negative entries in the extra row and column appropriately guarantees the desired bound on the length of the initial period, as we formalize in Section C.

# 5. Doubly exponential lower bound for fictitious play

We now turn to multi-player potential games. We will prove an exponentially stronger lower bound for fictitious play (FP).

Our construction is based on a connection we make to a graph-theoretic problem known as *snake in the box*. This was first introduced by Kautz (1958) in the context of coding theory, and has many interesting applications (Klee, 1970; 1967). The basic version of the problem is to identify a path along the edges of a high-dimensional hypercube with the following property. The path begins at some vertex of the hypercube and traverses the edges—two vertices are connected if they differ by a single bit—to as many vertices as it can reach, subject to the constraint that every time it arrives at a new vertex, the previous one and *all of its neighbors can never be used* going forward. Such a path is called a *snake*. Figure 3 (left) portrays an example for the 4-dimensional hypercube.

This problem can be defined for any graph. In our context, we associate each joint action with a vertex in the graph, and two vertices are connected with an edge if there is a unilateral deviation that goes from one to the other. Under this mapping, Figure 3 (right) shows a snake in a 3-player game where each player has 4 actions. Our key observation is that what essentially drove our lower bound in two-player games is the defining property of a snake. Indeed, the payoff matrix used in our two-player lower bound (Figure 2a) induces a snake that spirals toward the center.

For our purposes, it suffices to restrict our attention to an $n$-dimensional hypercube. A well-known fact is that there exists a snake with exponential length (Abbott & Katchalski, 1988; Evdokimov, 1969), although characterizing the precise length is a major open problem in graph theory.

**Lemma 5.1** (Evdokimov, 1969). *For any $n \in \mathbb{N}$, there exists a snake on the $n$-dimensional hypercube with length $C \cdot 2^n$, for some absolute constant $C > 0$.*

Let $P$ be such a snake in the $n$-dimensional hypercube. We proceed by embedding this path into a potential game. Specifically, we construct a binary-action $n$-player game with identical interests as follows. A joint action $(a_1, \ldots, a_n)$ is in one-to-one correspondence with vertices of the hypercube. For a joint action $\boldsymbol{a} = (a_1, \ldots, a_n) \in P$,

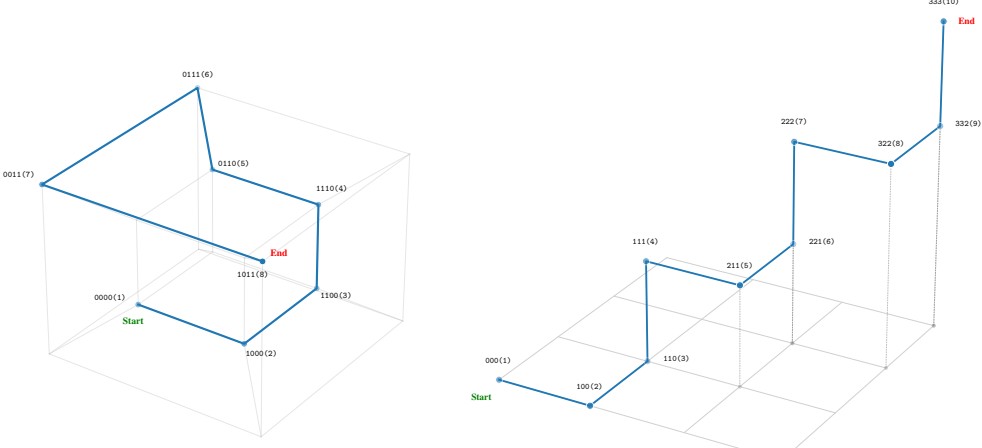

Figure 3. A snake on a 4-dimensional hypercube (left) and a snake corresponding to a 3-player 4-action game (right).

we let $p(\boldsymbol{a})$ be the position of $\boldsymbol{a}$ on the path $P$, starting from 1. We define the utility, for each player $i \in [n]$, as

$$u_i(\boldsymbol{a}) = \begin{cases} 0 & \text{if } \boldsymbol{a} \notin P, \\ p(\boldsymbol{a}) & \text{otherwise.} \end{cases} \quad (6)$$

We assume that FP is initialized at the start of the path $P$. By definition, it will always transition to pure strategies. Furthermore, because of the structure of the game, a basic invariance is that it will always remain on the path. Now, by construction, in every joint action on the path, there is a player who can improve the utility by at least 1 through a unilateral deviation, unless the joint action corresponds to the last vertex in the path, which is the global maximum of the utility function.

The crucial role of the snake property is this: at all iterations before reaching the end of the path, *exactly* one player will have positive best-response gap. This is because, by definition, all subsequent vertices except the immediate successor cannot be reached through a unilateral deviation. In other words, there are no shortcuts, and each edge traversal is caused by the movement of a single player.

If $T_k$ is the number of iterations that FP spent on the action profile corresponding to payoff $k$, we establish the following recursion (the proof is in Section F).

**Lemma 5.2.** *For any $k \geq 2$, $T_k \geq (k-1)T_{k-1} + T_{k-2} + T_{k-3} - \sum_{\kappa=1}^{k-4} \kappa T_\kappa$. ($T_k$ for $k \leq 0$ is to be interpreted as 0.)*

We draw attention to the fact that except the first three leading terms, the rest are negative. This is unlike the basic recursion that can be set up in two-player games, and happens because a player continually switches from one action to the other; in two-player games, a player always switches

to a new action, which has accumulated negative regret throughout.

We next argue that, despite the negative terms, $T_k \geq (k-1)!$ for $k \geq 1$.

**Lemma 5.3.** *Any sequence satisfying the recurrence relation of Lemma 5.2 grows at least as $(k-1)!$ for $k \geq 1$.*

Combining, we arrive at the following *doubly* exponential lower bound for FP in potential games.

**Theorem 5.4.** *FP requires at least $(C2^n)!$ to converge to a Nash equilibrium of an $n$-player potential game, where $C > 0$ is some absolute constant.*

This holds even for converging to a $\epsilon$-Nash equilibrium with $\epsilon < 1$, although the payoffs in our construction can be exponentially large. By rescaling—FP is scale invariant—it follows that doubly exponentially many iterations are needed to converge to an $\epsilon$-Nash equilibrium with $\epsilon \approx 1/2^n$. This closely matches our upper bound in Theorem 3.4.

Theorem 5.4 improves exponentially over existing lower bounds for fictitious play in potential games (Panageas et al., 2023; Brandt et al., 2013).

## 6. Conclusions and future research

We have established that FTRL, a celebrated online optimization algorithm, can take exponentially many iterations to converge to approximate Nash equilibria in potential games. We also showed a *doubly* exponential lower bound for fictitious play in multi-player games. An interesting question is whether a doubly exponential lower bound applies to FTRL when the learning rate is small; the primary challenge is that it is significantly harder to hide the path—

as we did for fictitious play—when players are mixing.

## Impact statement

This paper presents work whose goal is to advance the theoretical foundations of game theory and machine learning. We do not foresee any immediate societal consequences or ethical concerns.

## Acknowledgments

Ioannis Panageas is supported by NSF grant CCF-2454115. Tuomas Sandholm is supported by NIH award A240108S001, the Vannevar Bush Faculty Fellowship ONR N00014-23-1-2876, and National Science Foundation grant RI-2312342. Ioannis Anagnostides is grateful to Emanuel Tewolde and Brian Hu Zhang for helpful discussions at the initial stage of this project.

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

## A. Proofs from Section 3

This section contains the proofs omitted from Section 3. We begin with the one-step improvement property of FTRL.

**Lemma 3.1.** *If the sequence $(\boldsymbol{x}_i^{(t)})_{t \geq 1}$ is updated using FTRL under a 1-strongly convex regularizer $\mathcal{R}$ with respect to some norm $\|\cdot\|$,*

$$\langle \boldsymbol{x}_i^{(t+1)}, \boldsymbol{u}_i^{(t)} \rangle - \langle \boldsymbol{x}_i^{(t)}, \boldsymbol{u}_i^{(t)} \rangle \geq \frac{1}{\eta} \|\boldsymbol{x}_i^{(t+1)} - \boldsymbol{x}_i^{(t)}\|^2.$$

*Proof.* Let $\psi^{(t)}(\boldsymbol{x}_i) = \left\langle \boldsymbol{x}_i, \sum_{\tau=1}^{t} \boldsymbol{u}_i^{(\tau)} \right\rangle - \frac{1}{\eta}\mathcal{R}(\boldsymbol{x}_i)$. By definition, $\boldsymbol{x}_i^{(t+1)}$ maximizes $\psi^{(t)}(\boldsymbol{x}_i)$ with respect to $\mathcal{X}$. The first-order optimality condition implies

$$\langle \nabla \psi^{(t)}(\boldsymbol{x}_i^{(t+1)}), \boldsymbol{x}_i^{(t+1)} - \boldsymbol{x}_i' \rangle \geq 0 \quad \forall \boldsymbol{x}_i' \in \mathcal{X}_i.$$

Substituting the gradient $\nabla \psi^{(t)}(\boldsymbol{x}_i) = \sum_{\tau=1}^{t} \boldsymbol{u}_i^{(\tau)} - \frac{1}{\eta}\nabla\mathcal{R}(\boldsymbol{x}_i)$, we have

$$\left\langle \sum_{\tau=1}^{t} \boldsymbol{u}_i^{(\tau)} - \frac{1}{\eta}\nabla\mathcal{R}(\boldsymbol{x}_i^{(t+1)}), \boldsymbol{x}_i^{(t+1)} - \boldsymbol{x}_i' \right\rangle \geq 0 \quad \forall \boldsymbol{x}_i' \in \mathcal{X}_i. \tag{7}$$

Similarly, the optimality condition for $\boldsymbol{x}_i^{(t)}$ (which maximizes $\psi^{(t-1)}$) implies

$$\left\langle \sum_{\tau=1}^{t-1} \boldsymbol{u}_i^{(\tau)} - \frac{1}{\eta}\nabla\mathcal{R}(\boldsymbol{x}_i^{(t)}), \boldsymbol{x}_i^{(t)} - \boldsymbol{x}_i' \right\rangle \geq 0 \quad \forall \boldsymbol{x}_i' \in \mathcal{X}_i. \tag{8}$$

Substituting $\boldsymbol{x}_i' = \boldsymbol{x}_i^{(t)}$ in (7) and $\boldsymbol{x}_i' = \boldsymbol{x}_i^{(t+1)}$ in (8),

$$\left\langle \sum_{\tau=1}^{t} \boldsymbol{u}_i^{(\tau)} - \frac{1}{\eta}\nabla\mathcal{R}(\boldsymbol{x}_i^{(t+1)}), \boldsymbol{x}_i^{(t+1)} - \boldsymbol{x}_i^{(t)} \right\rangle \geq 0, \tag{9}$$

$$\left\langle \sum_{\tau=1}^{t-1} \boldsymbol{u}_i^{(\tau)} - \frac{1}{\eta}\nabla\mathcal{R}(\boldsymbol{x}_i^{(t)}), \boldsymbol{x}_i^{(t)} - \boldsymbol{x}_i^{(t+1)} \right\rangle \geq 0. \tag{10}$$

Summing (9) and (10) yields

$$\left\langle \sum_{\tau=1}^{t} \boldsymbol{u}_i^{(\tau)}, \boldsymbol{x}_i^{(t+1)} - \boldsymbol{x}_i^{(t)} \right\rangle + \left\langle \sum_{\tau=1}^{t-1} \boldsymbol{u}_i^{(\tau)}, \boldsymbol{x}_i^{(t)} - \boldsymbol{x}_i^{(t+1)} \right\rangle - \frac{1}{\eta}\left\langle \nabla\mathcal{R}(\boldsymbol{x}_i^{(t+1)}) - \nabla\mathcal{R}(\boldsymbol{x}_i^{(t)}), \boldsymbol{x}_i^{(t+1)} - \boldsymbol{x}_i^{(t)} \right\rangle \geq 0.$$

Since

$$\left\langle \sum_{\tau=1}^{t} \boldsymbol{u}_i^{(\tau)}, \boldsymbol{x}_i^{(t+1)} - \boldsymbol{x}_i^{(t)} \right\rangle + \left\langle \sum_{\tau=1}^{t-1} \boldsymbol{u}_i^{(\tau)}, \boldsymbol{x}_i^{(t)} - \boldsymbol{x}_i^{(t+1)} \right\rangle = \langle \boldsymbol{u}_i^{(t)}, \boldsymbol{x}_i^{(t+1)} - \boldsymbol{x}_i^{(t)} \rangle,$$

we have

$$\langle \boldsymbol{u}_i^{(t)}, \boldsymbol{x}_i^{(t+1)} - \boldsymbol{x}_i^{(t)} \rangle - \frac{1}{\eta}\left\langle \nabla\mathcal{R}(\boldsymbol{x}_i^{(t+1)}) - \nabla\mathcal{R}(\boldsymbol{x}_i^{(t)}), \boldsymbol{x}_i^{(t+1)} - \boldsymbol{x}_i^{(t)} \right\rangle \geq 0.$$

Since $\mathcal{R}$ is a 1-strongly convex function,

$$\left\langle \nabla\mathcal{R}(\boldsymbol{x}_i^{(t+1)}) - \nabla\mathcal{R}(\boldsymbol{x}_i^{(t)}), \boldsymbol{x}_i^{(t+1)} - \boldsymbol{x}_i^{(t)} \right\rangle \geq \|\boldsymbol{x}_i^{(t+1)} - \boldsymbol{x}_i^{(t)}\|^2,$$

and the claim follows. □

We next leverage Proposition 3.2 to establish that simultaneous FTRL converges to a limit set in which the potential has the same value. We refer to Losert & Akin (1983) for a related argument.

**Proposition A.1.** *In an $n$-player potential game, the $\omega$-limit set of simultaneous FTRL has the same potential value.*

*Proof.* By Proposition 3.2, we know that

$$\Phi(\boldsymbol{x}^{(t+1)}) - \Phi(\boldsymbol{x}^{(t)}) \geq \frac{1}{2\eta} \sum_{i=1}^{n} \|\boldsymbol{x}_i^{(t+1)} - \boldsymbol{x}_i^{(t)}\|^2 \geq 0.$$

Since $\Phi$ is bounded and nondecreasing along the trajectory, the sequence $\{\Phi(\boldsymbol{x}^{(t)})\}$ converges to a limit $\Phi^*$. Let $\Omega$ be the $\omega$-limit set—the set of accumulation points—of the sequence $\{\boldsymbol{x}^{(t)}\}$. Since $\mathcal{X}$ is compact, $\Omega$ is nonempty. By continuity, $\Phi(\boldsymbol{x}) = \Phi^*$ for all $\boldsymbol{x} \in \Omega$. Because the potential is constant on $\Omega$, the improvement at every step must be zero. By Lemma 3.1, this implies that any $\boldsymbol{x} \in \Omega$ is a *fixed point* of FTRL. $\square$

Furthermore, it is worth pointing out that if the limit point exists, it is a Nash equilibrium. Indeed, let us assume that $\boldsymbol{x} = \lim_{t\to\infty} \boldsymbol{x}^{(t)}$, so FTRL converges pointwise. For the sake of contradiction, suppose that $\boldsymbol{x}$ is not a Nash equilibrium. This means that there exists player $i$, deviation $\boldsymbol{x}_i^*$, and $\epsilon > 0$ such that

$$\langle \nabla_{\boldsymbol{x}_i}\Phi(\boldsymbol{x}), \boldsymbol{x}_i^* \rangle > \langle \nabla_{\boldsymbol{x}_i}\Phi(\boldsymbol{x}), \boldsymbol{x}_i \rangle + \epsilon. \tag{11}$$

Since $\Phi$ is continuously differentiable, $\lim_{t\to\infty} \boldsymbol{u}_i^{(t)} = \nabla_{\boldsymbol{x}_i}\Phi(\boldsymbol{x})$. This also implies $\lim_{t\to\infty} \frac{1}{t}\sum_{\tau=1}^{t} \boldsymbol{u}_i^{(\tau)} = \nabla_{\boldsymbol{x}_i}\Phi(\boldsymbol{x})$. Now, the FTRL update for player $i$ at time $t+1$ can be expressed as

$$\boldsymbol{x}_i^{(t+1)} = \arg\max_{\boldsymbol{x}_i' \in \mathcal{X}_i} \left( \left\langle \boldsymbol{x}_i', \frac{1}{t}\sum_{\tau=1}^{t} \boldsymbol{u}_i^{(\tau)} \right\rangle - \frac{1}{\eta t}\mathcal{R}_i(\boldsymbol{x}_i') \right).$$

By Berge's maximum theorem, this implies

$$\boldsymbol{x}_i = \lim_{t\to\infty} \boldsymbol{x}_i^{(t)} \in \arg\max_{\boldsymbol{x}' \in \mathcal{X}_i} \langle \boldsymbol{x}_i', \nabla_{\boldsymbol{x}_i}\Phi(\boldsymbol{x}) \rangle,$$

which is a contradiction to (11).

In fact, this argument connecting pointwise convergence to Nash equilibria is significantly more general, and holds for any no-regret algorithm and general-sum game.

We continue with the proof of Theorem 3.4, which is recalled below.

**Theorem 3.4.** *Alternating $\epsilon$-lazy no-regret dynamics converge to a $2\epsilon$-Nash equilibrium after at most $\exp\big(O_\epsilon(1/\epsilon^2)\big)$ rounds.*

*Proof.* We first point out that the total number of updates is bounded by $O_\epsilon(1/\epsilon)$. This follows directly from the potential property, the fact that the updates are alternating, and the definition of $\epsilon$-lazy updates.

**Lemma A.2** (Number of updates). *Let $K$ denote the total number of times an update occurs under $\epsilon$-lazy alternating no-regret dynamics, that is, $\boldsymbol{x}_i^{(t+1)} \neq \boldsymbol{x}_i^{(t)}$ for some player $i$. Then*

$$K \leq \frac{\Phi_{\text{range}}}{\epsilon}.$$

What remains is to bound the number of rounds in between two consecutive updates.

**Lemma A.3** (Time between updates). *Let $T_k$ be the time index of the $k$th update, which is assumed to be large enough so that $\text{Reg}_i^{(t)} \leq \frac{1}{2}\epsilon t$ for all $t \geq T_k$. Suppose that during $[T_k, T_{k+1} - 1]$, the strategy is fixed at $\boldsymbol{x}^{(T_k)}$ and is not an $2\epsilon$-Nash equilibrium. Then there exists a game-dependent parameter $P > 0$ such that*

$$T_{k+1} \leq T_k \left( 1 + \frac{P}{\epsilon} \right).$$

*Proof.* Let $t < T_{k+1}$. By definition of the no-regret property,

$$\sum_{\tau=1}^{t} \langle \boldsymbol{u}_i^{(\tau)}, \boldsymbol{x}_i' - \tilde{\boldsymbol{x}}_i^{(\tau)} \rangle \leq \text{Reg}_i^{(t)}.$$

Bounding the difference in the first $T_k$ rounds as $\sum_{\tau=1}^{T_k}\langle \boldsymbol{u}_i^{(\tau)}, \boldsymbol{x}_i' - \tilde{\boldsymbol{x}}_i^{(\tau)}\rangle \leq \sum_{\tau=1}^{T_k}\|\boldsymbol{x}_i' - \tilde{\boldsymbol{x}}_i^{(\tau)}\|\|\boldsymbol{u}_i^{(t)}\|_* \leq T_k BD_i$, we have

$$\sum_{\tau=T_k+1}^{t}\langle \boldsymbol{x}_i' - \tilde{\boldsymbol{x}}_i^{(\tau)}, \boldsymbol{u}_i^{(T_k)}\rangle \leq \mathsf{Reg}_i^{(t)} + T_k BD_i.$$

Here we used the fact that for $T_k < t < T_{k+1}$ no updates occur, so $\boldsymbol{u}_i^{(t)} = \boldsymbol{u}_i^{(T_k)}$. We now write

$$(t - T_k)\langle \boldsymbol{x}_i' - \boldsymbol{x}_i^{(T_k)}, \boldsymbol{u}_i^{(T_k)}\rangle + \sum_{\tau=T_k+1}^{t}\langle \boldsymbol{x}_i^{(T_k)} - \tilde{\boldsymbol{x}}_i^{(\tau)}, \boldsymbol{u}_i^{(T_k)}\rangle \leq T_k BD_i + \mathsf{Reg}_i^{(t)}.$$

Since $\boldsymbol{x}_i^{(\tau)}$ is not getting updated before $T_{k+1}$, it follows that $\langle \boldsymbol{x}_i^{(\tau)}, \boldsymbol{u}_i^{(T_k)}\rangle \geq \langle \tilde{\boldsymbol{x}}_i^{(\tau)}, \boldsymbol{u}_i^{(T_k)}\rangle - \epsilon$. Furthermore, since $\boldsymbol{x}_i^{(T_k)}$ is not a $2\epsilon$-Nash equilibrium, we have

$$\epsilon(t - T_k) \leq T_k BD_i + \mathsf{Reg}_i^{(t)}.$$

Using the fact that $\mathsf{Reg}_i^{(t)} \leq \frac{1}{2}\epsilon t$ and rearranging concludes the proof. $\qquad\square$

In other words, we have shown that

$$T_{k+1} \leq T_k\left(1 + O_\epsilon\left(\frac{1}{\epsilon}\right)\right).$$

So,

$$T_k \leq \left(1 + O_\epsilon\left(\frac{1}{\epsilon}\right)\right)^k \leq \exp\left(kO_\epsilon\left(\frac{1}{\epsilon}\right)\right).$$

Combining with Lemma A.2, the total number of iterations needed is $\exp\left(O_\epsilon\left(\frac{1}{\epsilon^2}\right)\right)$. $\qquad\square$

## B. Properties of the regularizer

This section establishes basic properties of permutation-invariant regularizers.

**Definition B.1** (Permutation invariance). A function $\mathcal{R}: \Delta_m \to \mathbb{R}$ is called *permutation invariant* if for every permutation $\pi: [m] \to [m]$ and every $\boldsymbol{x} \in \Delta_m$,

$$\mathcal{R}(\boldsymbol{x}) = \mathcal{R}(\pi(\boldsymbol{x})),$$

where $\pi(\boldsymbol{x})[i] := \boldsymbol{x}[\pi(i)]$.

**Lemma B.2** (Permutation-invariant function has uniform minimizer on the simplex). *Let $\mathcal{R}: \Delta_m \to \mathbb{R}$ be a convex, permutation-invariant function. Then the uniform distribution $\frac{1}{m}\mathbf{1}$ is a minimizer of $\mathcal{R}$ over $\Delta_m$. If $\mathcal{R}$ is strictly convex, $\frac{1}{m}\mathbf{1}$ is the unique minimizer.*

*Proof.* Let $\boldsymbol{x}^\star \in \arg\min_{\boldsymbol{x}\in\Delta_m} \mathcal{R}(\boldsymbol{x})$. By permutation invariance (Definition B.1), for any permutation $\pi$,

$$\mathcal{R}(\pi(\boldsymbol{x}^\star)) = \mathcal{R}(\boldsymbol{x}^\star),$$

so $\pi(\boldsymbol{x}^\star)$ is also a minimizer.

Let $\mathfrak{S}_m$ denote the set of all permutations of $\{1,\ldots,m\}$. By convexity of $\mathcal{R}$ and Jensen's inequality,

$$\mathcal{R}\left(\frac{1}{|\mathfrak{S}_m|}\sum_{\pi\in\mathfrak{S}_m}\pi(\boldsymbol{x}^\star)\right) \leq \frac{1}{|\mathfrak{S}_m|}\sum_{\pi\in\mathfrak{S}_m}\mathcal{R}(\pi(\boldsymbol{x}^\star)) = \mathcal{R}(\boldsymbol{x}^\star).$$

The average over all permutations of $\boldsymbol{x}^\star$—and more generally of any vector in $\Delta_m$—is the uniform vector $\frac{1}{m}\mathbf{1}$. To see this, define

$$\bar{\boldsymbol{x}} := \frac{1}{|\mathfrak{S}_m|}\sum_{\pi\in\mathfrak{S}_m}\pi(\boldsymbol{x}^\star),$$

and denote its $j$th coordinate by $\bar{\boldsymbol{x}}[j]$. Decompose $\mathfrak{S}_m$ as $\mathfrak{S}_m = \bigcup_{i=1}^{m} \mathfrak{S}_m^{(i)}$, where $\mathfrak{S}_m^{(i)}$ is the set of permutations mapping coordinate $j$ to $i$. Clearly, $|\mathfrak{S}_m^{(i)}| = (m-1)!$ for each $i \in [m]$. Hence,

$$\bar{\boldsymbol{x}}[j] = \frac{1}{|\mathfrak{S}_m|} \sum_{\pi \in \mathfrak{S}_m} \pi(\boldsymbol{x}^\star)[j] = \frac{1}{|\mathfrak{S}_m|} \sum_{i=1}^{m} |\mathfrak{S}_m^{(i)}| \, \boldsymbol{x}^\star[i]$$

$$= \frac{(m-1)!}{m!} \sum_{i=1}^{m} \boldsymbol{x}^\star[i] = \frac{1}{m}.$$

Consequently, $\bar{\boldsymbol{x}} = \frac{1}{m}\mathbf{1}$ and $\mathcal{R}(\bar{\boldsymbol{x}}) \leq \mathcal{R}(\boldsymbol{x}^\star)$, so $\bar{\boldsymbol{x}}$ is also a minimizer. If $\mathcal{R}$ is strictly convex, equality holds only if $\boldsymbol{x}^\star = \bar{\boldsymbol{x}} = \frac{1}{m}\mathbf{1}$. $\qquad\square$

**Lemma B.3** (FTRL with a permutation-invariant regularizer preserves order). *Let $\mathcal{R} : \Delta(\mathcal{A}_i) \to \mathbb{R}$ be a permutation-invariant, strictly convex, and differentiable regularizer, and let $\eta > 0$. Then, for any actions $a_i, a_i' \in \mathcal{A}_i$ and any round $t$, it holds that*

$$\sum_{\tau=1}^{t} \boldsymbol{u}_i^{(\tau)}[a_i] \geq \sum_{\tau=1}^{t} \boldsymbol{u}_i^{(\tau)}[a_i'] \quad \Longleftrightarrow \quad \boldsymbol{x}_i^{(t+1)}[a_i] \geq \boldsymbol{x}_i^{(t+1)}[a_i'].$$

*Proof.* Since $\mathcal{R}$ is strictly convex and differentiable, the FTRL optimizer $\boldsymbol{x}_i^{(t+1)}$ is unique and satisfies the first-order condition

$$\nabla \mathcal{R}(\boldsymbol{x}_i^{(t+1)}) = \eta \sum_{\tau=1}^{t} \boldsymbol{u}_i^{(\tau)} + \lambda \mathbf{1}$$

for some scalar $\lambda$. Hence, for any actions $a_i, a_i' \in \mathcal{A}_i$,

$$\frac{\partial \mathcal{R}(\boldsymbol{x}_i^{(t+1)})}{\partial \boldsymbol{x}_i[a_i]} - \frac{\partial \mathcal{R}(\boldsymbol{x}_i^{(t+1)})}{\partial \boldsymbol{x}_i[a_i']} = \eta \left( \sum_{\tau=1}^{t} \boldsymbol{u}_i^{(\tau)}[a_i] - \sum_{\tau=1}^{t} \boldsymbol{u}_i^{(\tau)}[a_i'] \right). \tag{12}$$

Suppose $\sum_{\tau=1}^{t} \boldsymbol{u}_i^{(\tau)}[a_i] \geq \sum_{\tau=1}^{t} \boldsymbol{u}_i^{(\tau)}[a_i']$. Then (12) gives

$$\frac{\partial \mathcal{R}(\boldsymbol{x}_i^{(t+1)})}{\partial \boldsymbol{x}_i[a_i]} \geq \frac{\partial \mathcal{R}(\boldsymbol{x}_i^{(t+1)})}{\partial \boldsymbol{x}_i[a_i']}. \tag{13}$$

We claim this implies

$$\boldsymbol{x}_i^{(t+1)}[a_i] \geq \boldsymbol{x}_i^{(t+1)}[a_i'].$$

For the sake of contradiction, assume $\boldsymbol{x}_i^{(t+1)}[a_i] < \boldsymbol{x}_i^{(t+1)}[a_i']$. Let $\pi$ be the permutation that swaps $a_i$ and $a_i'$, and set $\boldsymbol{y} := \pi(\boldsymbol{x}_i^{(t+1)})$. By permutation invariance, $\mathcal{R}(\boldsymbol{y}) = \mathcal{R}(\pi(\boldsymbol{x}_i^{(t+1)})) = \mathcal{R}(\boldsymbol{x}_i^{(t+1)})$; moreover, the gradient operator is linear, which implies that the corresponding partial derivatives are swapped:

$$\frac{\partial \mathcal{R}(\boldsymbol{y})}{\partial \boldsymbol{x}_i[a_i]} = \frac{\partial \mathcal{R}(\boldsymbol{x}_i^{(t+1)})}{\partial \boldsymbol{x}_i[a_i']}, \qquad \frac{\partial \mathcal{R}(\boldsymbol{y})}{\partial \boldsymbol{x}_i[a_i']} = \frac{\partial \mathcal{R}(\boldsymbol{x}_i^{(t+1)})}{\partial \boldsymbol{x}_i[a_i]}. \tag{14}$$

Since $\mathcal{R}$ is strictly convex, we have

$$\langle \nabla \mathcal{R}(\boldsymbol{x}_i^{(t+1)}) - \nabla \mathcal{R}(\boldsymbol{y}), \boldsymbol{x}_i^{(t+1)} - \boldsymbol{y} \rangle > 0 \quad \text{for } \boldsymbol{x}_i^{(t+1)} \neq \boldsymbol{y}. \tag{15}$$

By definition of $\boldsymbol{y}$, it differs from $\boldsymbol{x}_i^{(t+1)}$ only in the coordinates $a_i$ and $a_i'$. Therefore, using (14), the inner product in (15) reduces to

$$2 \left( \frac{\partial \mathcal{R}(\boldsymbol{x}_i^{(t+1)})}{\partial \boldsymbol{x}_i[a_i]} - \frac{\partial \mathcal{R}(\boldsymbol{x}_i^{(t+1)})}{\partial \boldsymbol{x}_i[a_i']} \right) \left( \boldsymbol{x}_i^{(t+1)}[a_i] - \boldsymbol{x}_i^{(t+1)}[a_i'] \right) > 0. \tag{16}$$

We note that the case $\boldsymbol{x} = \boldsymbol{y}$ is trivial, since the claim follows immediately from (16). Under our assumption $\boldsymbol{x}_i^{(t+1)}[a_i] - \boldsymbol{x}_i^{(t+1)}[a_i'] < 0$, the above inequality forces

$$\frac{\partial \mathcal{R}(\boldsymbol{x}_i^{(t+1)})}{\partial \boldsymbol{x}_i[a_i]} - \frac{\partial \mathcal{R}(\boldsymbol{x}_i^{(t+1)})}{\partial \boldsymbol{x}_i[a_i']} < 0,$$

contradicting (13). Therefore, $\boldsymbol{x}_i^{(t+1)}[a_i] \geq \boldsymbol{x}_i^{(t+1)}[a_i']$.

Conversely, suppose that $\boldsymbol{x}_i^{(t+1)}[a_i] \geq \boldsymbol{x}_i^{(t+1)}[a_i']$ but $\sum_{\tau=1}^{t} \boldsymbol{u}_i^{(\tau)}[a_i] < \sum_{\tau=1}^{t} \boldsymbol{u}_i^{(\tau)}[a_i']$. Then (12) implies

$$\frac{\partial \mathcal{R}(\boldsymbol{x}_i^{(t+1)})}{\partial \boldsymbol{x}_i[a_i]} < \frac{\partial \mathcal{R}(\boldsymbol{x}_i^{(t+1)})}{\partial \boldsymbol{x}_i[a_i']},$$

which, by the same argument as above, forces $\boldsymbol{x}_i^{(t+1)}[a_i] < \boldsymbol{x}_i^{(t+1)}[a_i']$, contradicting the assumption. Combining both directions establishes the equivalence. □

## C. Proofs from Section 4

We begin with the following simple but crucial observation. It shows that any pure strategy whose *cumulative utility* up to round $t$ is smaller than the maximal *cumulative utility* by at least a constant $\gamma$ is assigned, in round $t + 1$, a probability that is inversely proportional to $\eta^{(t)}\gamma$. To simplify the exposition, we introduce the *cumulative utility gap*, which we henceforth refer to as the *gap*.

**Definition C.1** (Cumulative utility gap). For a player $i$, the gap of an action $a_i \in \mathcal{A}_i$ at time $t$ is

$$\mathrm{Gap}_i^{(t)}[a_i] := \max_{a_i' \in \mathcal{A}_i} \sum_{\tau=1}^{t} \boldsymbol{u}_i^{(\tau)}[a_i'] - \sum_{\tau=1}^{t} \boldsymbol{u}_i^{(\tau)}[a_i].$$

**Lemma 4.1.** *Suppose that* $(\boldsymbol{x}_i^{(t)})_{t \geq 1}$ *is updated using* FTRL. *If* $\mathrm{Gap}_i^{(t)}[a_i] \geq \gamma > 0$ *for some action* $a_i \in \mathcal{A}_i$, *then*

$$\boldsymbol{x}_i^{(t+1)}[a_i] \leq \frac{R}{\eta^{(t)}\gamma}, \tag{5}$$

*where $R$ denotes the range of the regularizer.*

*Proof.* Define

$$\boldsymbol{x}_i' := \boldsymbol{x}_i^{(t+1)} - \boldsymbol{x}_i^{(t+1)}[a_i]\boldsymbol{e}_{a_i} + \boldsymbol{x}_i^{(t+1)}[a_i]\boldsymbol{e}_{a_i'},$$

where

$$a_i' \in \arg\max_{a \in \mathcal{A}_i} \sum_{\tau=1}^{t} \boldsymbol{u}_i^{(\tau)}[a],$$

and $\boldsymbol{e}_a$ denotes the $a$th standard basis vector. Equivalently, $\boldsymbol{x}_i'$ differs from $\boldsymbol{x}_i^{(t+1)}$ only in the coordinates $a_i$ and $a_i'$, with $\boldsymbol{x}_i'[a_i] = 0$ and $\boldsymbol{x}_i'[a_i'] = \boldsymbol{x}_i^{(t+1)}[a_i] + \boldsymbol{x}_i^{(t+1)}[a_i']$. In particular, $\boldsymbol{x}_i' \in \Delta(\mathcal{A}_i)$.

Since $\mathrm{Gap}_i^{(t)}[a_i] \geq \gamma$, we obtain

$$\left\langle \boldsymbol{x}_i', \sum_{\tau=1}^{t} \boldsymbol{u}_i^{(\tau)} \right\rangle - \left\langle \boldsymbol{x}_i^{(t+1)}, \sum_{\tau=1}^{t} \boldsymbol{u}_i^{(\tau)} \right\rangle = \left\langle \boldsymbol{x}_i' - \boldsymbol{x}_i^{(t+1)}, \sum_{\tau=1}^{t} \boldsymbol{u}_i^{(\tau)} \right\rangle$$

$$= \boldsymbol{x}_i^{(t+1)}[a_i] \left( \sum_{\tau=1}^{t} \boldsymbol{u}_i^{(\tau)}[a_i'] - \sum_{\tau=1}^{t} \boldsymbol{u}_i^{(\tau)}[a_i] \right)$$

$$= \boldsymbol{x}_i^{(t+1)}[a_i] \, \mathrm{Gap}_i^{(t)}[a_i]$$

$$\geq \boldsymbol{x}_i^{(t+1)}[a_i] \, \gamma. \tag{17}$$

Moreover, the regularizer has bounded range, so $\mathcal{R}(\boldsymbol{x}_i') \leq \mathcal{R}(\boldsymbol{x}_i^{(t+1)}) + R$. On the other hand, $\boldsymbol{x}_i^{(t+1)}$ is the `FTRL` maximizer of the objective $\langle \boldsymbol{x}_i, \sum_{\tau=1}^{t} \boldsymbol{u}_i^{(\tau)} \rangle - \frac{1}{\eta^{(t)}} \mathcal{R}(\boldsymbol{x}_i)$, and therefore its value is at least that of feasible $\boldsymbol{x}_i'$:

$$\left\langle \boldsymbol{x}_i^{(t+1)}, \sum_{\tau=1}^{t} \boldsymbol{u}_i^{(\tau)} \right\rangle - \frac{1}{\eta^{(t)}} \mathcal{R}(\boldsymbol{x}_i^{(t+1)}) \geq \left\langle \boldsymbol{x}_i', \sum_{\tau=1}^{t} \boldsymbol{u}_i^{(\tau)} \right\rangle - \frac{1}{\eta^{(t)}} \mathcal{R}(\boldsymbol{x}_i'). \tag{18}$$

Combining (17) and (18) with the bound on $\mathcal{R}(\boldsymbol{x}_i')$ yields

$$\boldsymbol{x}_i^{(t+1)}[a_i] \leq \frac{R}{\eta^{(t)} \gamma},$$

which implies the claimed bound. $\qquad\square$

### C.1. Definition of transition period

As explained in the main body, we partition the `FTRL` dynamics into *periods*. A period $k$ is a block of consecutive rounds during which `FTRL` concentrates its probability mass on a small set of strategies—the *competing actions*—while all remaining actions receive negligible mass. We fix the threshold $\delta := \frac{1}{4m}$ and define periods accordingly. We now recall the basic setup.

For $k \geq 3$, let $t_k$ and $\overline{t_k}$ denote the first and last rounds of period $k$, respectively, and let $T_k := \overline{t_k} - t_k + 1$ denote its length, so that period $k$ corresponds to the interval $t \in [t_k, \overline{t_k}]$.

**Definition 4.2** (Transition period). For each integer $k \in \{3, 4, \ldots, 2m-1\}$, the *transition period* $k$ is the set of rounds $t$ for which the following conditions hold:

(i) If $k \geq 4$ is even, then for all $t \in [t_k, \overline{t_k}]$, $\boldsymbol{x}_2^{(t)}[a_2(k)] \geq 1 - \delta$ and $\boldsymbol{x}_1^{(t)}[a_1(k-1)] + \boldsymbol{x}_1^{(t)}[a_1(k)] \geq 1 - \delta$.

(ii) If $k \geq 5$ is odd, then for all $t \in [t_k, \overline{t_k}]$, $\boldsymbol{x}_1^{(t)}[a_1(k)] \geq 1 - \delta$ and $\boldsymbol{x}_2^{(t)}[a_2(k-1)] + \boldsymbol{x}_2^{(t)}[a_2(k)] \geq 1 - \delta$.

Since the actions $a_1(2)$ and $a_2(2)$ are undefined, the above definition does not apply to $k = 3$. We therefore let period 3 consist only of the first round, *i.e.*, $t_3 = \overline{t_3} = 1$, and let period 4 start immediately thereafter. The remainder of the analysis relies on Property 4.3, which we prove by induction in Section C.6.

**Property 4.3** (Period consistency). *Suppose both players employ `FTRL`. If a round $t$ belongs to period $k$ for some $3 \leq k \leq 2m-1$, then the subsequent round $t+1$ belongs either to the same period $k$ or to the next period $k+1$.*

### C.2. Uniform initialization

We begin with the initialization, which serves as the base case for the induction in the proof of Property 4.3. A standard initialization for `FTRL` is the uniform strategy. In particular, for permutation-invariant regularizers, the first update ($t = 1$) selects a minimizer of $\mathcal{R}$, which is the uniform distribution. Indeed, at initialization the cumulative utility gaps satisfy

$$\texttt{Gap}_i^{(0)}[a_i] = 0 \quad \text{for all } i \in \{1, 2\} \text{ and } a_i \in \mathcal{A}_i.$$

Thus, at $t = 1$ the `FTRL` update reduces to minimizing $\mathcal{R}$; by Lemma B.2, we obtain

$$\boldsymbol{x}_1^{(1)} = \text{Uniform}(\mathcal{A}_1), \qquad \boldsymbol{x}_2^{(1)} = \text{Uniform}(\mathcal{A}_2).$$

As we mentioned in Section 4, our construction builds on the class of games introduced by Panageas et al. (2023). For completeness, Section D reviews their recursive construction of the matrix $\mathbf{B}_{m,r}$ (*cf.* (64) in Definition D.1) and establishes the structural properties we require. Although this class is well suited for proving exponential lower bounds for fictitious play, it is not directly applicable to `FTRL`: under uniform initialization, running `FTRL` on $(\mathbf{B}_{m,r}, \mathbf{B}_{m,r})$ for any $r$ quickly identifies the actions attaining the maximum payoff and converges to that profile within a constant number of rounds. To preclude this behavior, we modify the payoff matrix as described below.

$$\begin{bmatrix} 3 & 0 & 0 & 0 \\ 0 & 7 & 0 & 6 \\ 0 & 8 & 9 & 0 \\ 4 & 0 & 0 & 5 \end{bmatrix}$$

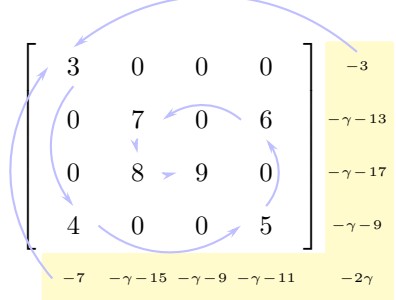

(a) Original matrix $\mathbf{B} = \mathbf{B}_{4,2}$ of size $4 \times 4$.

(b) New matrix $\mathbf{A}$ of size $5 \times 5$.

The underlying idea is straightforward. By Lemma 4.1, the gap $\mathtt{Gap}_1^{(1)}[a_1]$ (resp. $\mathtt{Gap}_2^{(1)}[a_2]$) at round 1 controls the probability assigned to action $a_1$ (resp. $a_2$) at round 2. Hence, by ensuring that a sufficiently large gap is incurred for all actions except $(1, 1)$ after round 1, we can guarantee that at round 2 the only action profile played with probability nearly 1 (up to $\delta$) is $(1, 1)$.

For an odd dimension $m$, we consider the matrix $\mathbf{B} = \mathbf{B}_{m-1,2}$ of size $(m - 1) \times (m - 1)$ as defined in Definition D.1. Starting from the matrix $\mathbf{B}$, we embed it into a larger matrix

$$\mathbf{A} \in [-\gamma - (4m - 1),\, 2m - 1]^{m \times m},$$

which is obtained by adding one extra row and one extra column to $\mathbf{A}$. The transformation is illustrated in Figures 4a and 4b.

$$\mathbf{A}[a_1, a_2] = \begin{cases} \mathbf{B}[a_1, a_2], & 1 \leq a_1 \leq m - 1 \text{ and } 1 \leq a_2 \leq m - 1, \\ -\sum_{a_1'=1}^{m-1} \mathbf{B}[a_1', 1], & a_1 = m \text{ and } a_2 = 1, \\ -\gamma - \sum_{a_1'=1}^{m-1} \mathbf{B}[a_1', a_2], & a_1 = m \text{ and } 2 \leq a_2 \leq m - 1, \\ -\sum_{a_2'=1}^{m-1} \mathbf{B}[1, a_2'], & a_2 = m \text{ and } a_1 = 1, \\ -\gamma - \sum_{a_2'=1}^{m-1} \mathbf{B}[a_1, a_2'], & a_2 = m \text{ and } 2 \leq a_1 \leq m - 1, \\ -2\,\gamma, & a_1 = m \text{ and } a_2 = m, \\ 0, & \text{otherwise.} \end{cases} \tag{19}$$

**Lemma C.2** (Structural properties of $\mathbf{A}$). *The following properties hold for $\mathbf{A}$:*

*(i) The positive entries $\{3, 4, \ldots, 2m - 1\}$ each appear exactly once.*

*(ii) $\max_{i,j} \mathbf{A}[i, j] = (2m - 1)$.*

*(iii) For even $k \in \{3, 4, \ldots, 2m - 2\}$, $a_1(k) = a_1(k + 1)$.*

*(iv) For odd $k \in \{3, 4, \ldots, 2m - 2\}$, $a_2(k) = a_2(k + 1)$.*

*Proof.* All claims follow from the properties established in Section D for the embedded submatrix $\mathbf{B}_{m-1,2}$ of $\mathbf{A}$. In particular, Item (i) is an immediate consequence of Lemma D.2. Moreover, by Item (i) in Proposition D.4, the maximum entry of $\mathbf{B}_{m-1,2}$ equals $2 + \big(2(m - 1) - 1\big) = 2m - 1$. proving Item (ii). Since the embedding preserves the values of $\mathbf{B}_{m-1,2}$, and all additional entries in $\mathbf{A}$ (in particular those in the extra row and column) are chosen to be negative, this entry is also the maximum of $\mathbf{A}$. The remaining properties follow directly from Items (ii) and (iii) in Proposition D.4, applied to the same embedded block. □

As illustrated below, this construction ensures that, under uniform initialization, all actions other than $(1,1)$ start with a utility gap of order $\Omega(\gamma/m)$, thereby enforcing that the action profile $(1,1)$ is played with probability nearly 1. The remainder of the analysis relies on Property 4.3, which we prove by induction in Section C.6. In what follows, we establish Property C.3, which will be used throughout the analysis.

**Property C.3.** *At round* 1*, the utility gaps satisfy*

$$\mathrm{Gap}_1^{(1)}[a_1] \geq \gamma^{(1)} \quad \text{and} \quad \mathrm{Gap}_2^{(1)}[a_2] \geq \gamma^{(1)},$$

*for all actions $a_1 \in \mathcal{A}_1$ and $a_2 \in \mathcal{A}_2$ except for action 1, which has zero gap, and $\gamma^{(1)}$ satisfies*

$$\gamma^{(1)} := \max\left\{ \frac{4m^3 R}{1}, 2\left(\frac{64Rm}{\delta}\right)^{\frac{1}{1-\alpha}}, \left(\frac{Rm}{\delta}\right)^{\frac{1}{1-\alpha}} \alpha^{\frac{\alpha}{1-\alpha}}(1-\alpha) \right\} \tag{20}$$

*Then, it also holds*

$$\boldsymbol{x}_2^{(2)}[a_2(4)] \geq 1-\delta, \quad \boldsymbol{x}_1^{(2)}[a_1(3)] + \boldsymbol{x}_1^{(2)}[a_1(4)] \geq 1-\delta, \quad \text{where } \delta = \frac{1}{4m}.$$

*In other words, period $k = 4$ starts at round* 2*, satisfying the Definition 4.2.*

**Lemma C.4.** *Let $\boldsymbol{x}_1^{(1)} = \boldsymbol{x}_2^{(1)} = \mathrm{Uniform}(m)$. If both players employ* FTRL *on the game $(\mathbf{A}, \mathbf{A})$, then Property C.3 is satisfied.*

*Proof.* From Lemma B.2, the FTRL update at round 1 dictates that both players choose the uniform strategy over their action sets, *i.e.*,

$$\boldsymbol{x}_1^{(1)} = \boldsymbol{x}_2^{(1)} = \mathrm{Uniform}(m).$$

Although the effect of (19) on the utility vectors $\boldsymbol{u}_1^{(1)}, \boldsymbol{u}_2^{(1)}$ is apparent, we present a detailed derivation for completeness. We describe Player 1's utility vector; Player 2's is analogous.

1. **Row $a_1 = 1$**

$$\begin{aligned}
\boldsymbol{u}_1^{(1)}[1] &:= \sum_{a_2=1}^{m} \boldsymbol{x}_2^{(1)}[a_2]\, \mathbf{A}^T[1, a_2] \\
&= \frac{1}{m}\left( \sum_{a_2=1}^{m-1} \mathbf{A}[1, a_2] + \mathbf{A}[1, m] \right) \\
&= \frac{1}{m}\left( \sum_{a_2=1}^{m} \mathbf{B}[1, a_2] + \left( -\sum_{a_2'=1}^{m} \mathbf{B}[1, a_2'] \right) \right) \\
&= 0.
\end{aligned}$$

2. **Rows $2 \leq a_1 \leq m-1$**

$$\begin{aligned}
\boldsymbol{u}_1^{(1)}[a_1] &:= \sum_{a_2=1}^{m} \boldsymbol{x}_2^{(1)}[a_2]\, \mathbf{A}[a_1, a_2] \\
&= \frac{1}{m}\left( \sum_{a_2=1}^{m-1} \mathbf{A}[a_1, a_2] + \mathbf{A}[a_1, m] \right) \\
&= \frac{1}{m}\left( \sum_{a_2=1}^{m-1} \mathbf{B}[a_1, a_2] + \left( -\gamma - \sum_{a_2'=1}^{m-1} \mathbf{B}[a_1, a_2'] \right) \right) \\
&= -\frac{\gamma}{m}.
\end{aligned}$$

3. **Row** $a_1 = m$

$$
\begin{aligned}
\boldsymbol{u}_1^{(1)}[m] &:= \sum_{a_2=1}^{m} \boldsymbol{x}_2^{(1)}[a_2]\, \mathbf{A}[m, a_2] \\
&= \frac{1}{m}\left(\mathbf{A}[m,1] + \sum_{a_2=2}^{m-1}\mathbf{A}[m,a_2] + \mathbf{A}[m,m]\right) \\
&= \frac{1}{m}\left(-\sum_{a_1'=1}^{m-1}\mathbf{B}[a_1',1] + \sum_{a_2=2}^{m-1}\left(-\gamma - \sum_{a_1'=1}^{m-1}\mathbf{B}[a_1',a_2]\right) - 2\gamma\right) \\
&= \frac{1}{m}\left(-(m-2)\gamma - 2\gamma - \sum_{a_1'=1}^{m-1}\mathbf{B}[a_1',1] - \sum_{a_2=2}^{m-1}\sum_{a_1'=1}^{m-1}\mathbf{B}[a_1',a_2]\right) \\
&= \frac{1}{m}\left(-m\gamma - \sum_{a_1=1}^{m-1}\sum_{a_2=1}^{m-1}\mathbf{B}[a_1,a_2]\right) \\
&= -\gamma - \frac{1}{m}\sum_{a_1=1}^{m-1}\sum_{a_2=1}^{m-1}\mathbf{B}[a_1,a_2].
\end{aligned}
$$

Only action 1 has non-negative utility at round 1, while all other actions have utility gaps of order $\Omega\left(\gamma/m\right)$, as shown below. Analogously, the same holds for Player 2's utility vector $\boldsymbol{u}_2^{(1)}$. Since $\gamma$ is a free parameter that can be chosen arbitrarily large, Lemma 4.1 and this construction ensure that under uniform initialization, the action profile $(1,1)$ will be played at round 2 with probability nearly 1.

$$
\mathtt{Gap}_1^{(1)}[a_1] = \begin{cases} 0 & a_1 = 1, \\ \frac{\gamma}{m} & 2 \leq a_1 \leq m-1, \\ \gamma + \frac{\sum_{a_1,a_2}\mathbf{B}[a_1,a_2]}{m} & a_1 = m. \end{cases}
$$

$$
\mathtt{Gap}_2^{(1)}[a_2] = \begin{cases} 0 & a_2 = 1, \\ \frac{\gamma}{m} & 2 \leq a_2 \leq m-1, \\ \gamma + \frac{\sum_{a_1,a_2}\mathbf{B}[a_1,a_2]}{m} & a_2 = m. \end{cases}
$$

Setting $\gamma^{(1)} := \frac{\gamma}{m}$, it follows that $\mathtt{Gap}_i^{(1)}[a] \geq \gamma^{(1)}$ for every $a \neq 1$. Now, by Lemma 4.1 and setting $\gamma^{(1)} = \frac{4m^3 R}{1}$ it follows that for all $a_1 \neq 1$,

$$
\boldsymbol{x}_1^{(2)}[a_1] \; \leq \; \frac{R}{\eta^{(1)}\mathtt{Gap}_1^{(1)}[a_1]} \; \leq \; \frac{R}{\left(\frac{1}{1^\alpha}\right)\left(\frac{4m^3 R}{1} \cdot \frac{1}{m}\right)} \; = \; \frac{1}{4m^2} \; \leq \; \frac{1}{4m}\frac{1}{m} \; = \; \frac{\delta}{m}.
$$

and therefore

$$
\boldsymbol{x}_1^{(2)}[1] \; = \; 1 - \sum_{a_1=2}^{m}\boldsymbol{x}_1^{(2)}[a_1] \; \geq \; 1 - \frac{m}{4m^2} \; \geq \; 1 - \frac{1}{4m} \; = \; 1 - \delta.
$$

An analogous argument applies to Player 2's strategy $\boldsymbol{x}_2^{(1)}$. Therefore, in accordance with Definition 4.2, period $k = 3$ starts at round 1 and terminates immediately, so period 4 begins at round 2. Indeed, at round 2 the defining conditions hold:

$$
\boldsymbol{x}_1^{(2)}[a_1(3)] + \boldsymbol{x}_1^{(2)}[a_1(4)] \geq 1 - \delta, \qquad \boldsymbol{x}_2^{(2)}[a_2(4)] \geq 1 - \delta.
$$

Finally, setting

$$
\gamma^{(1)} \; = \; \max\left\{\frac{4m^3 R}{1}, 2\left(\frac{64Rm}{\delta}\right)^{\frac{1}{1-\alpha}}, \left(\frac{Rm}{\delta}\right)^{\frac{1}{1-\alpha}}\alpha^{\frac{\alpha}{1-\alpha}}(1-\alpha)\right\}, \quad \text{where} \quad \delta = \frac{1}{4m}.
$$

ensures that Property C.3 is satisfied. □

*Remark* C.5 (Normalized payoff matrix $\mathbf{A}$). Our lower bound construction can be implemented with payoffs in $[-1, 1]$. Although FTRL is not scale invariant—rescaling payoffs changes the updates—we can modify the construction so that it does not rely on large-magnitude entries.

Specifically, assume henceforth that payoffs must lie in $[-1, 1]$. We start from the recursive matrix $\mathbf{B}_{m-1,2}$ in Definition D.1. By Item (i) in Proposition D.4, its maximum entry equals $2m - 1$. Define the rescaled matrix $\mathbf{B} := \frac{1}{2m-1} \mathbf{B}_{m-1,2}$, so that $\mathbf{B} \in [0, 1]^{(m-1) \times (m-1)}$. We then construct our payoff matrix $\mathbf{A}$ by embedding $\mathbf{B}$, as in the original construction, and padding it with $\lceil \gamma^{(1)} \rceil$ additional *rows* and *columns*, where $\gamma^{(1)}$ is as in (20). All newly introduced entries are set to $-1$, except that in each added row and each added column we set the first entry to $0$. This should be contrasted with the initial construction in (19), which adds only one row and one column but uses entries of magnitude $\Theta(\gamma^{(1)})$. Since $\lceil \gamma^{(1)} \rceil = \mathsf{poly}(m)$, the resulting game has polynomial dimension and all payoffs lie in $[-1, 1]$. It is easy to verify that this modification does not affect the subsequent arguments; consequently, the same lower bound holds: FTRL still requires $2^{\Omega(\mathsf{poly}(m))}$ rounds.

## C.3. Growth of the cumulative utility gaps

As shown in Lemma 4.1, to control the probability assigned to an action it suffices to track its *utility gap* over time, rather than the full cumulative utility vector.

**Lemma C.6.** *Let $k \geq 4$ be even. For any action $a_1 \in \mathcal{A}_1 \setminus \{a_1(k), a_1(k-2)\}$, the utility gap $\mathsf{Gap}_1^{(t)}[a_1]$ increases by at least $(1 - \delta)(k - 1) - \delta(2m - 1)$ at every round $t \in [\underline{t_k}, \overline{t_k}]$. Similarly, if $k \geq 5$ is odd, then for any action $a_2 \in \mathcal{A}_2 \setminus \{a_2(k), a_2(k-2)\}$, the utility gap $\mathsf{Gap}_2^{(t)}[a_2]$ increases by at least $(1 - \delta)(k - 1) - \delta(2m - 1)$ at every round $t \in [\underline{t_k}, \overline{t_k}]$.*

*Proof.* Let $a_1 \in \mathcal{A}_1 \setminus \{a_1(k), a_1(k-1)\}$ for even $k$. By the definition of period $k$, we have

$$\boldsymbol{x}_1^{(t)}[a_1(k-1)] + \boldsymbol{x}_1^{(t)}[a_1(k)] \geq 1 - \delta, \qquad \boldsymbol{x}_2^{(t)}[a_2(k)] \geq 1 - \delta.$$

Since during period $k$ Player 1 primarily mixes between the actions $a_1(k-1)$ and $a_1(k)$, it follows from Lemma E.1 that either $\boldsymbol{x}_1^{(t)}[a_1(k-1)]$ or $\boldsymbol{x}_1^{(t)}[a_1(k)]$ is maximal among the components of $\boldsymbol{x}_1^{(t)}$. By Lemma B.3, a maximizer of the cumulative utility at round $t - 1$ also belongs to that set. Consequently,

$$\{a_1(k-1), a_1(k)\} \supseteq \arg\max_{a_1' \in \mathcal{A}_1} \left\{ \sum_{\tau=1}^{t-1} \boldsymbol{u}_1^{(\tau)}[a_1'] \right\} \neq \emptyset.$$

Since $k$ is even, Item (iv) in Lemma C.2 implies $a_2(k-1) = a_2(k)$, and so the utilities of actions $a_1(k-1)$ and $a_1(k)$ at round $t$ satisfy

$$\boldsymbol{u}_1^{(t)}[a_1(k-1)] \geq \boldsymbol{x}_2^{(t)}[a_2(k-1)](k-1) \geq (1-\delta)(k-1), \qquad \boldsymbol{u}_1^{(t)}[a_1(k)] \geq \boldsymbol{x}_2^{(t)}[a_2(k)]k \geq (1-\delta)k. \quad (21)$$

Next, consider the one-step change in the gap of action $a_1$ after round $t$:

$$
\begin{aligned}
\Delta\mathsf{Gap}_1^{(t)}[a_1] &:= \mathsf{Gap}_1^{(t)}[a_1] - \mathsf{Gap}_1^{(t-1)}[a_1] \\
&= \left( \max_{a_1' \in \mathcal{A}_1} \sum_{\tau=1}^{t} \boldsymbol{u}_1^{(\tau)}[a_1'] - \sum_{\tau=1}^{t} \boldsymbol{u}_1^{(\tau)}[a_1] \right) - \left( \max_{a_1' \in \mathcal{A}_1} \sum_{\tau=1}^{t-1} \boldsymbol{u}_1^{(\tau)}[a_1'] - \sum_{\tau=1}^{t-1} \boldsymbol{u}_1^{(\tau)}[a_1] \right) \\
&= \left( \max_{a_1' \in \mathcal{A}_1} \sum_{\tau=1}^{t} \boldsymbol{u}_1^{(\tau)}[a_1'] - \max_{a_1' \in \mathcal{A}_1} \sum_{\tau=1}^{t-1} \boldsymbol{u}_1^{(\tau)}[a_1'] \right) - \boldsymbol{u}_1^{(t)}[a_1]. \quad (22)
\end{aligned}
$$

Even though (during period $k$) one of $a_1(k-1)$ and $a_1(k)$ attains the maximal cumulative utility at rounds $t - 1$ and $t$, the maximizer could in principle change from one round to the next. Fortunately, Lemma E.2 avoids a case distinction.

$$\max_{a_1' \in \mathcal{A}_1} \sum_{\tau=1}^{t} \boldsymbol{u}_1^{(\tau)}[a_1'] - \max_{a_1' \in \mathcal{A}_1} \sum_{\tau=1}^{t-1} \boldsymbol{u}_1^{(\tau)}[a_1'] \geq \min\left\{ \boldsymbol{u}_1^{(t)}[a_1(k-1)], \boldsymbol{u}_1^{(t)}[a_1(k)] \right\}. \quad (23)$$

Combining (22), (23), and (21), we obtain

$$\Delta \mathrm{Gap}_1^{(t)}[a_1] \geq (1-\delta)(k-1) - \boldsymbol{u}_1^{(t)}[a_1].$$

For any $a_1 \in \mathcal{A}_1 \setminus \{a_1(k), a_1(k-1)\}$, the identity $a_2(k-1) = a_2(k)$ (by Item (iv) in Lemma C.2) implies $\mathbf{A}[a_1, a_2(k)] = 0$; equivalently, row $a_1$ has no nonzero entry in column $a_2(k)$. Moreover, by the definition of period $k$, Player 2 assigns total probability at most $\delta$ to columns other than $a_2(k)$. Finally, Item (ii) in Lemma C.2 implies that every payoff outside column $a_2(k)$ is at most $2m - 1$. Hence,

$$\boldsymbol{u}_1^{(t)}[a_1] \leq \delta(2m-1).$$

Therefore, due to $a_1(k-2) = a_1(k-1)$ from Item (iii) in Lemma C.2, it follows that for any $a_1 \in \mathcal{A}_1 \setminus \{a_1(k), a_1(k-1)\} = \mathcal{A}_1 \setminus \{a_1(k), a_1(k-2)\}$,

$$\Delta \mathrm{Gap}_1^{(t)}[a_1] \geq (1-\delta)(k-1) - \delta(2m-1).$$

The proof for odd $k$ and for Player 2 is analogous.

$\square$

**Lemma C.7.** *Let $k \geq 4$ be even. For any action $a_2 \in \mathcal{A}_2 \setminus \{a_2(k), a_2(k+2)\}$, the utility gap $\mathrm{Gap}_2^{(t)}[a_2]$ increases by at least $(1-\delta)(k-1) - \delta(2m-1)$ at every round $t \in [\underline{t_k}, \overline{t_k}]$. Similarly, if $k \geq 5$ is odd, then for any action $a_1 \in \mathcal{A}_1 \setminus \{a_1(k), a_1(k+2)\}$, the utility gap $\mathrm{Gap}_1^{(t)}[a_1]$ increases by at least $(1-\delta)(k-1) - \delta(2m-1)$ at every round $t \in [\underline{t_k}, \overline{t_k}]$.*

*Proof.* Let $a_2 \in \mathcal{A}_2 \setminus \{a_2(k), a_2(k+2)\}$ and suppose that $k$ is even. By the definition of transition period $k$, for every $t \in [\underline{t_k}, \overline{t_k}]$,

$$\boldsymbol{x}_1^{(t)}[a_1(k-1)] + \boldsymbol{x}_1^{(t)}[a_1(k)] \geq 1 - \delta, \qquad \boldsymbol{x}_2^{(t)}[a_2(k)] \geq 1 - \delta.$$

Since $\boldsymbol{x}_2^{(t)}[a_2(k)] \geq 1 - \delta$, action $a_2(k)$ has maximal probability among actions in $\mathcal{A}_2$, and thus Lemma B.3 implies that it also attains the maximal cumulative utility up to time $t - 1$, *i.e.*,

$$a_2(k) \in \arg \max_{a_2' \in \mathcal{A}_2} \sum_{\tau=1}^{t-1} \boldsymbol{u}_2^{(\tau)}[a_2'].$$

Consequently, the one-step change in the gap of $a_2$ at round $t$ satisfies

$$\begin{aligned}
\Delta \mathrm{Gap}_2^{(t)}[a_2] &:= \mathrm{Gap}_2^{(t)}[a_2] - \mathrm{Gap}_2^{(t-1)}[a_2] \\
&= \left( \max_{a_2' \in \mathcal{A}_2} \sum_{\tau=1}^{t} \boldsymbol{u}_2^{(\tau)}[a_2'] - \max_{a_2' \in \mathcal{A}_2} \sum_{\tau=1}^{t-1} \boldsymbol{u}_2^{(\tau)}[a_2'] \right) - \boldsymbol{u}_2^{(t)}[a_2] \\
&\geq \left( \sum_{\tau=1}^{t} \boldsymbol{u}_2^{(\tau)}[a_2(k)] - \sum_{\tau=1}^{t-1} \boldsymbol{u}_2^{(\tau)}[a_2(k)] \right) - \boldsymbol{u}_2^{(t)}[a_2] \\
&= \boldsymbol{u}_2^{(t)}[a_2(k)] - \boldsymbol{u}_2^{(t)}[a_2]. \tag{24}
\end{aligned}$$

For the first term, the defining condition of period $k$ yields $\boldsymbol{u}_2^{(t)}[a_2(k)] = \boldsymbol{x}_1^{(t)}[a_1(k)] \, k + \boldsymbol{x}_1^{(t)}[a_1(k-1)] \, (k-1) \geq (1-\delta)(k-1)$. For the second term, the identities $a_2(k-1) = a_2(k)$ and $a_2(k+1) = a_2(k+2)$ (by Item (iv) in Lemma C.2) imply that, for any $a_2 \in \mathcal{A}_2 \setminus \{a_2(k), a_2(k+2)\}$, $\mathbf{A}[a_1(k-1), a_2] = 0$ and $\mathbf{A}[a_1(k), a_2] = 0$. Moreover, by the definition of period $k$, Player 1 assigns total probability at most $\delta$ to actions other than $a_1(k-1)$ and $a_1(k)$, and Item (ii) in Lemma C.2 implies that every payoff outside these two rows is at most $2m - 1$. Therefore,

$$\boldsymbol{u}_2^{(t)}[a_2] \leq \delta(2m-1). \tag{25}$$

Combining (24) and (25), we conclude that for any $a_2 \in \mathcal{A}_2 \setminus \{a_2(k), a_2(k+2)\}$,

$$\Delta \mathrm{Gap}_2^{(t)}[a_2] = \boldsymbol{u}_2^{(t)}[a_2(k)] - \boldsymbol{u}_2^{(t)}[a_2] \geq (1-\delta)(k-1) - \delta(2m-1).$$

This establishes the claim. The proof for odd $k$ (and the corresponding case for Player 1) is analogous. $\square$

### C.4. Lower bounds on period $k$ duration

So far, we have characterized how the gap of an action $a_i \in \mathcal{A}_i$ evolves over the rounds within period $k$. We now use this characterization to bound the duration of period $k$.

**Lemma C.8.** *Let $k \geq 4$ be even. Then*

$$T_k \geq \frac{1}{1 + \delta(2m - 1)} \, Gap_1^{(\overline{t_{k-1}})}[a_1(k)].$$

*Similarly, if $k \geq 5$ is odd, then*

$$T_k \geq \frac{1}{1 + \delta(2m - 1)} \, Gap_2^{(\overline{t_{k-1}})}[a_2(k)].$$

*Proof.* Let $k$ be even (the odd-$k$ case is symmetric). By definition, period $k$ ends at the round $t$ when $\boldsymbol{x}_1^{(t)}[a_1(k)]$ reaches the threshold $1 - \delta$ (and, for odd $k$, once the probability mass on $\boldsymbol{x}_2^{(t)}[a_2(k)]$ reaches $1 - \delta$). Before this happen, the action $a_1(k)$ needs first to become a maximizer of the cumulative utility over $\mathcal{A}_1$ (as shown in Property 4.3) ; this occurs at round $\hat{t}_k \in [\underline{t_k}, \overline{t_k}]$.

For every $t \in [\underline{t_k}, \hat{t}_k - 1]$, the one-step change in the utility gap of $a_1(k)$ is lower bounded by

$$
\begin{aligned}
\Delta Gap_1^{(t)}[a_1(k)] &= \left( \max_{a_1' \in \mathcal{A}_1} \sum_{\tau=1}^{t} \boldsymbol{u}_1^{(\tau)}[a_1'] - \max_{a_1' \in \mathcal{A}_1} \sum_{\tau=1}^{t-1} \boldsymbol{u}_1^{(\tau)}[a_1'] \right) - \boldsymbol{u}_1^{(t)}[a_1(k)] \\
&= \left( \sum_{\tau=1}^{t} \boldsymbol{u}_1^{(\tau)}[a_1(k-1)] - \sum_{\tau=1}^{t-1} \boldsymbol{u}_1^{(\tau)}[a_1(k-1)] \right) - \boldsymbol{u}_1^{(t)}[a_1(k)] \\
&= \boldsymbol{u}_1^{(t)}[a_1(k-1)] - \boldsymbol{u}_1^{(t)}[a_1(k)] \\
&\geq \boldsymbol{x}_2^{(t)}[a_2(k-1)](k-1) - \boldsymbol{x}_2^{(t)}[a_2(k)] \, k - \boldsymbol{x}_2^{(t)}[a_2(k+1)](k+1).
\end{aligned}
$$

Item (iv) in Lemma C.2 implies that $a_2(k-1) = a_2(k)$ for $k$ even, and so

$$
\begin{aligned}
\Delta Gap_1^{(t)}[a_1(k)] &\geq -\boldsymbol{x}_2^{(t)}[a_2(k)] - \boldsymbol{x}_2^{(t)}[a_2(k+1)](k+1) \\
&= -\boldsymbol{x}_2^{(t)}[a_2(k)] - \boldsymbol{x}_2^{(t)}[a_2(k+1)](2m - 1) \\
&\geq -1 - \delta(2m - 1),
\end{aligned}
$$

where the last inequality uses $\boldsymbol{x}_2^{(t)}[a_2(k+1)] \leq \delta$ and $\boldsymbol{x}_2^{(t)}[a_2(k)] \leq 1$. Equivalently, over period $k$ the gap $Gap_1^{(t)}[a_1(k)]$ can decrease by at most $1 + \delta(2m - 1)$ per round. Moreover, since period $k$ lasts at least until $\hat{t}_k$, we have $T_k \geq (\hat{t}_k - 1) - \underline{t_k} + 1$. Because the gap at the beginning of the period is finite, it follows that

$$T_k \geq \frac{1}{1 + \delta(2m - 1)} \, Gap_1^{(\overline{t_{k-1}})}[a_1(k)],$$

which establishes the claim. The odd-$k$ case follows analogously.

$\square$

**Lemma 4.4.** *Let $k \geq 6$ be even. Then*

$$Gap_1^{(\overline{t_{k-2}})}[a_1(k)] \geq \sum_{\ell=4}^{k-2} \left[ (1 - \delta)(\ell - 1) - \delta(2m - 1) \right] T_\ell.$$

*Similarly, if $k \geq 5$ is odd, then*

$$Gap_2^{(\overline{t_{k-2}})}[a_2(k)] \geq \sum_{\ell=4}^{k-2} \left[ (1 - \delta)(\ell - 1) - \delta(2m - 1) \right] T_\ell.$$

*Proof.* The claim follows by repeatedly applying Lemma C.6. We present the argument for even $k \geq 6$; the odd-$k$ case is analogous.

For any period index $\ell \in \{4, \ldots, k-2\}$, applying Lemma C.6 within period $\ell$ yields that the regret gap of action $a_1(k)$ increases by at least

$$\Big[(1-\delta)(\ell-1) - \delta(2m-1)\Big]$$

per round throughout that period. Therefore, over the entire duration $T_\ell$ of period $\ell$, we obtain the increment bound

$$\mathrm{Gap}_1^{(\overline{t_\ell})}[a_1(k)] \geq \mathrm{Gap}_1^{(\overline{t_{\ell-1}})}[a_1(k)] + \Big[(1-\delta)(\ell-1) - \delta(2m-1)\Big] T_\ell.$$

Summing the above inequality over $\ell = 4, \ldots, k-2$ and telescoping gives

$$\mathrm{Gap}_1^{(\overline{t_{k-2}})}[a_1(k)] \geq \mathrm{Gap}_1^{(\overline{t_3})}[a_1(k)] + \sum_{\ell=4}^{k-2}\Big[(1-\delta)(\ell-1) - \delta(2m-1)\Big] T_\ell.$$

Dropping the nonnegative term $\mathrm{Gap}_1^{(\overline{t_3})}[a_1(k)] = \mathrm{Gap}_1^{(1)}[a_1(k)] \geq 0$ results to the stated lower bound:

$$\mathrm{Gap}_1^{(\overline{t_{k-2}})}[a_1(k)] \geq \sum_{\ell=4}^{k-2}\Big[(1-\delta)(\ell-1) - \delta(2m-1)\Big] T_\ell.$$

The odd-$k$ case follows analogously. $\qquad\square$

Although Lemmas C.8 and 4.4 appear to admit a recursive application, a mismatch in their reference points prevents this directly. Specifically, Lemma 4.4 provides a lower bound on the regret gap only up to the end of period $k-2$, whereas Lemma C.8 concerns the subsequent period $k-1$. To bridge this gap, we require an auxiliary lemma that tracks the evolution of the regret gap of $a_1(k)$ from the end of $k-2$ through the end of $k-1$.

**Lemma C.9.** *Let $k \geq 4$ be even. Then,*

$$\mathrm{Gap}_1^{(\overline{t_{k-1}})}[a_1(k)] \geq \mathrm{Gap}_1^{(\overline{t_{k-2}})}[a_1(k)] - (1 + \delta(2m-1)) T_{k-1}.$$

*Similarly, if $k \geq 5$ is odd, then*

$$\mathrm{Gap}_2^{(\overline{t_{k-1}})}[a_2(k)] \geq \mathrm{Gap}_2^{(\overline{t_{k-2}})}[a_2(k)] - (1 + \delta(2m-1)) T_{k-1}.$$

*Proof.* We track the evolution of $\mathrm{Gap}_1^{(t)}[a_1(k)]$ during period $k-1$ for even $k$. Fix any $t \in [t_{k-1}, \overline{t_{k-1}}]$. By the definition of transition period $k-1$, it holds that

$$\boldsymbol{x}_2^{(t)}[a_2(k-2)] + \boldsymbol{x}_2^{(t)}[a_2(k-1)] \geq 1 - \delta \quad \text{and} \quad \boldsymbol{x}_1^{(t)}[a_1(k-1)] \geq 1 - \delta.$$

Since Player 1 assigns probability at least $1 - \delta$ to $a_1(k-1)$, Lemma B.3 implies that $a_1(k-1)$ attains the maximal cumulative utility in $\mathcal{A}_1$ throughout period $k-1$. Therefore, the maximizer in the definition of the gap is $a_1(k-1)$ and

$$\begin{aligned}
\Delta\mathrm{Gap}_1^{(t)}[a_1(k)] &:= \mathrm{Gap}_1^{(t)}[a_1(k)] - \mathrm{Gap}_1^{(t-1)}[a_1(k)] \\
&= \left(\max_{a_1' \in \mathcal{A}_1} \sum_{\tau=1}^{t} \boldsymbol{u}_1^{(\tau)}[a_1'] - \max_{a_1' \in \mathcal{A}_1} \sum_{\tau=1}^{t-1} \boldsymbol{u}_1^{(\tau)}[a_1']\right) - \boldsymbol{u}_1^{(t)}[a_1(k)] \\
&= \boldsymbol{u}_1^{(t)}[a_1(k-1)] - \boldsymbol{u}_1^{(t)}[a_1(k)].
\end{aligned}$$

Expanding the utilities gives

$$\Delta\mathrm{Gap}_1^{(t)}[a_1(k)] = \boldsymbol{x}_2^{(t)}[a_2(k-1)](k-1) + \boldsymbol{x}_2^{(t)}[a_2(k-2)](k-2) - \boldsymbol{x}_2^{(t)}[a_2(k)] k - \boldsymbol{x}_2^{(t)}[a_2(k+1)](k+1).$$

Since $k$ is even, Item (iv) in Lemma C.2 implies $a_2(k-1) = a_2(k)$. We now derive a crude but sufficient bound. The terms satisfy $\boldsymbol{x}_2^{(t)}[a_2(k-2)] \geq 0$ and $\boldsymbol{x}_2^{(t)}[a_2(k-1)] \leq 1$. Moreover, by the definition of the period, Player 2 assigns probability at most $\delta$ to any action other than $a_2(k-2), a_2(k-1)$, hence $\boldsymbol{x}_2^{(t)}[a_2(k+1)] \leq \delta$. Substituting these bounds yields

$$\Delta \mathrm{Gap}_1^{(t)}[a_1(k)] \geq (k-1) - k - \delta(k+1) = -1 - \delta(k+1) \geq -1 - \delta(2m-1),$$

where the last inequality uses $k+1 \leq 2m-1$. Thus, throughout period $k-1$, the gap $\mathrm{Gap}_1^{(t)}[a_1(k)]$ can decrease by at most $1 + \delta(2m-1)$ per round, and therefore

$$\mathrm{Gap}_1^{(\overline{t_{k-1}})}[a_1(k)] \geq \mathrm{Gap}_1^{(\overline{t_{k-2}})}[a_1(k)] - \left(1 + \delta(2m-1)\right) T_{k-1}.$$

The odd-$k$ case follows by an analogous argument. $\qquad \square$

## C.5. Exponential lower bound

Here we combine the lemmas from the preceding subsection to obtain an exponential lower bound. In particular, we show that the combined duration of periods $2m-4$ and $2m-3$ is already exponential in $m$.

**Lemma C.10** (Lower bound on $T_4$). *Under Property C.3, it holds that*

$$T_4 \geq \frac{\gamma_1}{2}.$$

*Proof.* By Lemma C.8, for $k=4$ (even) we have $T_4 \geq \frac{1}{1+\delta(2m-1)} \mathrm{Gap}_1^{(\overline{t_3})}[a_1(4)]$. By Property C.3, period 3 is the single round 1, so $\mathrm{Gap}_1^{(\overline{t_3})}[a_1(4)] = \mathrm{Gap}_1^{(1)}[a_1(4)] \geq \gamma^{(1)}$. Substituting yields

$$T_4 \geq \frac{\gamma_1}{1+\delta(2m-1)}.$$

Finally, since $\delta = \frac{1}{4m}$, it follows that $1 + \delta(2m+1) \leq 1 + \frac{2m-1}{4m} \leq 2$, and therefore $T_4 \geq \gamma_1/2$, as claimed. $\qquad \square$

**Lemma 4.5** (Recurrence relation of $T_k + T_{k-1}$). *For a small enough $\delta > 0$,*

$$T_k + T_{k-1} \geq \frac{1}{2} \sum_{\ell=4}^{k-2} (\ell - 2) T_\ell$$

*for all $k \geq 6$.*

*Proof.* We present the argument for even $k$; the case of odd $k$ is analogous. By Lemma C.8, the length of period $k$ satisfies

$$T_k \geq \frac{1}{1+\delta(2m-1)} \mathrm{Gap}_1^{(\overline{t_{k-1}})}[a_1(k)]. \tag{26}$$

To lower bound the gap at time $\overline{t_{k-1}}$, we first invoke Lemma 4.4, which yields

$$\mathrm{Gap}_1^{(\overline{t_{k-2}})}[a_1(k)] \geq \sum_{\ell=4}^{k-2} \left[(1-\delta)(\ell-1) - \delta(2m-1)\right] T_\ell. \tag{27}$$

Next, to relate the gap at times $\overline{t_{k-2}}$ and $\overline{t_{k-1}}$, we apply Lemma C.9, obtaining

$$\mathrm{Gap}_1^{(\overline{t_{k-1}})}[a_1(k)] \geq \mathrm{Gap}_1^{(\overline{t_{k-2}})}[a_1(k)] - \left(1 + \delta(2m-1)\right) T_{k-1}. \tag{28}$$

Substituting (27) into (28) and then into (26) gives

$$T_k \geq \frac{\sum_{\ell=4}^{k-2} \left[(1-\delta)(\ell-1) - \delta(2m-1)\right] T_\ell}{1+\delta(2m-1)} - T_{k-1}. \tag{29}$$

Rearranging (29) yields

$$T_k + T_{k-1} \geq \frac{\sum_{\ell=4}^{k-2} \left[ (1-\delta)(\ell-1) - \delta(2m-1) \right] T_\ell}{1 + \delta(2m-1)}. \tag{30}$$

By Lemma E.3, for $\ell \geq 4$ we have

$$(1-\delta)(\ell-1) - \delta(2m-1) \geq \ell - 2.$$

Moreover, since $\delta = \frac{1}{4m}$ (from Property C.3), we have

$$1 + \delta(2m-1) \leq 1 + \frac{2m-1}{4m} \leq 2.$$

Plugging these bounds into (30) proves that claim

$$T_k + T_{k-1} \geq \frac{1}{2} \sum_{\ell=4}^{k-2} (\ell-2) \, T_\ell.$$

This completes the proof for even $k$; the odd-$k$ case follows analogously. $\qquad\square$

**Lemma C.11** (Exponential lower bound). *Under Property C.3, it holds that*

$$T_{2m-3} + T_{2m-4} \geq \frac{\gamma^{(1)}}{4} \left( \frac{m-3}{e} \right)^{m-3}.$$

*Proof.* Define, for $3 \leq k \leq m$,

$$S_k := T_{2k-1} + T_{2k-2}, \qquad P_k := \sum_{\ell'=3}^{k} (\ell'-2) \, S_{\ell'}.$$

Applying Lemma 4.5 with index $2k-1$ gives

$$S_k = T_{2k-1} + T_{2k-2} \geq \frac{1}{2} \sum_{\ell=4}^{2k-3} (\ell-2) \, T_\ell.$$

Grouping the terms $(T_{2\ell-2}, T_{2\ell-1})$ for $\ell = 4, \ldots, k-1$ and using that all coefficients are nonnegative yields a lower bound

$$\frac{1}{2} \sum_{\ell=4}^{2k-3} (\ell-2) \, T_\ell \geq \frac{1}{2} \sum_{\ell=3}^{k-1} ((2\ell-2)-2) \, (T_{2\ell-1} + T_{2\ell-2}) = \sum_{\ell'=3}^{k-1} (\ell'-2) \, S_{\ell'} = P_{k-1}.$$

Therefore, for all $k \geq 4$,

$$S_k \geq P_{k-1}. \tag{31}$$

It follows that for every $k \geq 4$,

$$P_k = P_{k-1} + (k-2) S_k \geq P_{k-1} + (k-2) P_{k-1} = (k-1) P_{k-1},$$

where we used (31). Iterating from $k = 4$ up to $k = m-2$ yields

$$P_{m-1} \geq \left( \prod_{j=4}^{m-2} (j-1) \right) P_3 = \frac{(m-3)!}{2} P_3 = \frac{(m-3)!}{2} S_3,$$

since $P_3 = (3-2) S_3 = S_3$. Finally, (31) with $k = m-1$ gives $S_{m-1} \geq P_{m-2}$, and $S_{m-1} = T_{2m-3} + T_{2m-4}$ by definition, so

$$T_{2m-3} + T_{2m-4} = S_{m-1} \geq P_{m-2} \geq \frac{(m-3)!}{2} S_3.$$

Moreover, $S_3 = T_5 + T_4 \geq T_4$, and Lemma C.10 gives $T_4 \geq \gamma^{(1)}/2$, proving the factorial bound.

Applying Stirling's lower bound $(m-3)! \geq \left(\frac{m-3}{e}\right)^{m-3}$ yields

$$T_{2m-3} + T_{2m-4} \;\geq\; \frac{\gamma^{(1)}}{4} \left(\frac{m-3}{e}\right)^{m-3}.$$

$\square$

## C.6. Period consistency

Here, we justify the definition of a *period* (Definition 4.2) and establish its consistency. To this end, we prove two lemmas. Throughout, we focus on the case where $k$ is even; the case of odd $k$ follows by an entirely analogous argument.

We show that the FTRL dynamics are consistent with the definition of a period. In particular, if round $t$ belongs to period $k$, then round $t+1$ should belong only to period $k$, unless $\boldsymbol{x}_1^{(t+1)}[a_1(k)] \geq 1 - \delta$, in which case period $k$ terminates at round $t$. Hence, we have to rule out *period mixing*: since FTRL induces mixed strategies, it is not a priori clear that the update cannot move directly from period $k$ to some period other than $k+1$. To exclude this, we proceed in two steps. First, Lemma C.12 shows that any action whose gap grows linearly in $t$ receives at most $\delta/m$ probability mass under the next FTRL update. We then leverage this fact in Property 4.3 to preclude transitions to periods other than $k+1$.

**Lemma C.12** (Uniformly small probability under linear gap growth)**.** *Let $a_i \in \mathcal{A}_i$ and suppose that action $a_i$ increases its gap by at least $1$ in each round, i.e., $\mathrm{Gap}_i^{(t+1)}[a_i] \geq \mathrm{Gap}_i^{(t)}[a_i] + 1$ for $t \geq 1$. Then, it holds*

$$\boldsymbol{x}_i^{(t+1)}[a_i] \leq \frac{R}{\eta^{(t)} \, \mathrm{Gap}_i^{(t)}[a_i]} \leq \frac{\delta}{m}, \quad \textit{for every } t \geq 1 \,. \tag{32}$$

*Proof.* The proof relies only on Lemma 4.1, which implies that for every $t \geq 1$,

$$\boldsymbol{x}_i^{(t+1)}[a_i] \;\leq\; \frac{R}{\eta^{(t)} \, \mathrm{Gap}_i^{(t)}[a_i]}.$$

By (20) in Property C.3, the initial gap $\gamma^{(1)}$ satisfies $\mathrm{Gap}_i^{(1)}[a_i] = \gamma^{(1)} \geq \frac{Rm}{\delta}$, which ensures the desired bound on $\boldsymbol{x}_i^{(2)}[a_i]$ at round 2. To control all subsequent rounds under $\eta^{(t)} = t^{-\alpha}$, it is enough to guarantee that

$$\mathrm{Gap}_i^{(t)}[a_i] \;\geq\; \frac{Rm}{\delta} t^\alpha \qquad \forall t \geq 1, \tag{33}$$

since this immediately implies $\boldsymbol{x}^{(t+1)}[a_i] \leq \delta/m$ for all $t \geq 1$. To establish (33), we use the assumed linear growth of the gap: $\mathrm{Gap}_i^{(t)}[a_i] \geq \mathrm{Gap}_i^{(1)}[a_i] + (t-1) = \gamma^{(1)} + (t-1)$ for all $t \geq 1$. Thus, a sufficient condition for (33) is $\gamma^{(1)} + (t-1) \geq \frac{Rm}{\delta} t^\alpha$ for all $t \geq 1$. Although the linear term on the left eventually dominates the sublinear term on the right, we need the bound to hold uniformly from the start; this is ensured by $\gamma^{(1)}$ from (20) in Property C.3. Since $t - 1 \leq t$ for all $t \geq 1$, it suffices to require

$$\gamma^{(1)} \;\geq\; \frac{Rm}{\delta} t^\alpha - t \qquad \forall t \geq 1. \tag{34}$$

The tightest sufficient condition in (34) is obtained by maximizing the right-hand side over $t \geq 1$. Let $c := Rm/\delta$ and define $f(t) := ct^\alpha - t$ for $t \geq 1$. We view $t$ as a continuous variable and upper bound $\sup_{t \geq 1} f(t)$ by maximizing $f$ over $t \in [1, \infty)$. Since $\alpha \in [0, 1)$, $f$ is concave on $(0, \infty)$ and hence attains its maximum at a unique critical point. Differentiating yields $f'(t) = c\alpha t^{\alpha - 1} - 1$, so the maximizer satisfies $t^\star = (c\alpha)^{\frac{1}{1-\alpha}}$. Evaluating $f$ at $t^\star$, we obtain

$$f(t^\star) = c(t^\star)^\alpha - t^\star = c(c\alpha)^{\frac{\alpha}{1-\alpha}} - (c\alpha)^{\frac{1}{1-\alpha}} = \left(c^{\frac{1}{1-\alpha}}\right) \alpha^{\frac{\alpha}{1-\alpha}} (1 - \alpha)$$

Define $\gamma^\star := f(t^\star)$. Then, by (20) in Property C.3, we have $\gamma^{(1)} \geq \gamma^\star$, and therefore (34) holds. This implies (33), and consequently

$$\boldsymbol{x}_i^{(t+1)}[a_i] \leq \frac{\delta}{m} \qquad \text{for all } t \geq 1.$$

$\square$

**Property 4.3** (Period consistency). *Suppose both players employ* FTRL. *If a round $t$ belongs to period $k$ for some $3 \leq k \leq 2m - 1$, then the subsequent round $t + 1$ belongs either to the same period $k$ or to the next period $k + 1$.*

*Proof.* Assume that $k$ is even. We prove, by induction on the period index, that the FTRL dynamics satisfy the conditions of Definition 4.2. For notational convenience, for each $k \in \{3, \ldots, 2m - 1\}$, define the tail action sets $\mathcal{A}_1(k) := \{a_1(k') : k \leq k' \leq 2m - 1\}$ and $\mathcal{A}_2(k) := \{a_2(k') : k \leq k' \leq 2m - 1\}$.

By Definition 4.2, period 3 consists of the single round 1, *i.e.*, $\underline{t_3} = \overline{t_3} = 1$. This case is already verified and is included purely by convention. At round 2, Property C.3 shows us

$$x_1^{(2)}[a_1(3)] + x_1^{(2)}[a_1(4)] \geq 1 - \delta, \qquad x_2^{(2)}[a_2(4)] \geq 1 - \delta. \tag{35}$$

Hence, period 4 begins at round $\underline{t_4} = 2$, and the defining conditions are satisfied at its start.

Assume now that periods $3, 4, \ldots, k - 1$ satisfy Definition 4.2, and consider the initial round $\underline{t_k}$ of period $k$. By Definition 4.2, we have

$$x_2^{(\underline{t_k})}[a_2(k)] \geq 1 - \delta, \tag{36}$$

$$x_1^{(\underline{t_k})}[a_1(k-1)] + x_1^{(\underline{t_k})}[a_1(k)] \geq 1 - \delta. \tag{37}$$

It therefore suffices to show that the period-$k$ conditions continue to hold at every round $t \in \{\underline{t_k}, \underline{t_k} + 1, \ldots, \overline{t_k}\}$. Our main tools are Lemma C.12 and the gap-to-probability bound Lemma 4.1.

**Player 1.** By Lemma C.6 and Lemma E.3, every action $a_1 \in \mathcal{A}_1(k + 2)$ increases its gap by at least $(k - 2)$ at each round $t \in [1, \underline{t_k}]$. Consequently, Lemma C.12 implies that every $a_1 \in \mathcal{A}_1(k + 2)$ receives negligible probability mass at the next update:

$$x_1^{(\underline{t_k}+1)}[a_1] \leq \frac{\delta}{m}.$$

Next, consider actions $a_1 \in [m] \setminus \mathcal{A}_1(k - 2)$. Although these actions may have been active in earlier periods, they have remained *inactive* (*i.e.*, assigned probability at most $\delta/m$) throughout the last two periods, $(k - 2)$ and $(k - 1)$. By Lemma C.6 and Lemma E.3, each such $a_1$ increases its gap by at least $(k - 2)$ per round over the interval $[\underline{t_{k-2}}, \overline{t_{k-1}}]$. Moreover, Lemma C.8 ensures that the length of this interval is linear in the elapsed horizon:

$$T_{k-1} + T_{k-2} \geq \frac{1}{4} \sum_{\ell=3}^{k-1} T_\ell = \frac{1}{4} \overline{t_{k-1}}.$$

Therefore, the cumulative gap of any $a_1 \in [m] \setminus \mathcal{A}_1(k - 2)$ is still linear in time, and another application of Lemma C.12 yields the uniform bound

$$x_1^{(\underline{t_k}+1)}[a_1] \leq \frac{\delta}{m}.$$

It remains to control the evolution of the competing actions $a_1(k - 1)$ and $a_1(k)$. By the definition of the gap, we have

$$\Delta\mathrm{Gap}_1^{(\underline{t_k})}[a_1(k)] := u_1^{(\underline{t_k})}[a_1(k-1)] - u_1^{(\underline{t_k})}[a_1(k)]$$
$$= \left( x_2^{(\underline{t_k})}[a_2(k-1)](k-1) + x_2^{(\underline{t_k})}[a_2(k-2)](k-2) \right)$$
$$- \left( x_2^{(\underline{t_k})}[a_2(k)]k + x_2^{(\underline{t_k})}[a_2(k+1)](k+1) \right). \tag{38}$$

Since $k$ is even, Item (iv) in Lemma C.2 implies $a_2(k - 1) = a_2(k)$, and therefore (38) simplifies to

$$\Delta\mathrm{Gap}_1^{(\underline{t_k})}[a_1(k)] = -x_2^{(\underline{t_k})}[a_2(k)] - x_2^{(\underline{t_k})}[a_2(k+1)](k+1) + x_2^{(\underline{t_k})}[a_2(k-2)](k-2). \tag{39}$$

By (36), $x_2^{(\underline{t_k})}[a_2(k)] \geq 1 - \delta$, so the total probability mass on columns other than $a_2(k)$ is at most $\delta$. Using (39) and dropping the negative term $-x_2^{(\underline{t_k})}[a_2(k+1)](k+1)$, we obtain

$$\Delta\mathrm{Gap}_1^{(\underline{t_k})}[a_1(k)] \leq -x_2^{(\underline{t_k})}[a_2(k)] + \delta(k-2)$$
$$\leq -(1 - \delta) + \delta(k-2) = -1 + \delta(k-1).$$

Under the bounds $k \leq 2m + 1$ and $\delta = \frac{1}{4m}$, we have $\delta(k-1) \leq \frac{1}{2}$, and therefore

$$\Delta\mathrm{Gap}_1^{(t_k)}[a_1(k)] \; \leq \; -\frac{1}{2}. \tag{40}$$

In particular, as long as $\boldsymbol{x}_2^{(t_k)}[a_2(k)] \geq 1 - \delta$, the gap of $a_1(k)$ decreases by at least $1/2$ per round at time $\underline{t_k}$.

**Player 2.** The same reasoning applies symmetrically to Player 2. In particular, every action in $\mathcal{A}_2 \setminus \{a_2(k), a_2(k+1)\}$ continues to have probability at most $\delta/m$ by another application of Lemma C.12. The argument is identical to the one for Player 1 after splitting the actions into (i) the *future* actions $\mathcal{A}_2(k+2)$ and (ii) the actions *active* in *earlier* periods, namely $[m] \setminus \mathcal{A}_2(k-2)$.

The remaining nontrivial case concerns the competing actions $a_2(k)$ and $a_2(k+1)$. Since $a_2(k)$ has the largest probability mass, Lemma B.3 implies that it also attains the maximal cumulative utility among actions in $\mathcal{A}_2$. Thus, the one-step update of $\mathrm{Gap}^{(t_k)}[a_2(k+1)]$ is

$$
\begin{aligned}
\Delta\mathrm{Gap}^{(t_k)}[a_2(k+1)] &:= \boldsymbol{u}_2^{(t_k)}[a_2(k)] - \boldsymbol{u}_2^{(t_k)}[a_2(k+1)] \\
&= \boldsymbol{x}_1^{(t_k)}[a_1(k)]k + \boldsymbol{x}_1^{(t_k)}[a_1(k-1)](k-1) \\
&\quad - \boldsymbol{x}_1^{(t_k)}[a_1(k+1)](k+1) - \boldsymbol{x}_1^{(t_k)}[a_1(k+2)](k+2).
\end{aligned}
$$

Since $k$ is even, Item (iii) in Lemma C.2 implies $a_1(k) = a_1(k+1)$, and therefore

$$\Delta\mathrm{Gap}^{(t_k)}[a_2(k+1)] = -\boldsymbol{x}_1^{(t_k)}[a_1(k)] + \boldsymbol{x}_1^{(t_k)}[a_1(k-1)](k-1) - \boldsymbol{x}_1^{(t_k)}[a_1(k+2)](k+2). \tag{41}$$

The sign of $\Delta\mathrm{Gap}^{(t_k)}[a_2(k+1)]$ is not immediate, as it depends on the relative magnitudes of $\boldsymbol{x}_1^{(t_k)}[a_1(k)]$ and $\boldsymbol{x}_1^{(t_k)}[a_1(k-1)]$, while $\boldsymbol{x}_1^{(t_k)}[a_1(k+2)]$ is uniformly small. Nevertheless, we can extract a meaningful lower bound. As long as $\boldsymbol{x}_1^{(t_k)}[a_1(k-1)] \geq \boldsymbol{x}_1^{(t_k)}[a_1(k)]$ (which holds at round $\underline{t_k}$ by the definition of the period), (37) implies

$$\boldsymbol{x}_1^{(t_k)}[a_1(k-1)] \; \geq \; \frac{1}{2}(1-\delta).$$

Using this in (41) and the bound $\boldsymbol{x}_1^{(t_k)}[a_1(k+2)] \leq \delta$ yields

$$\Delta\mathrm{Gap}^{(t_k)}[a_2(k+1)] \; \geq \; \frac{1}{2}(1-\delta)(k-2) - \delta(k+2).$$

Under our standing choice of parameters and Lemma E.3, the right-hand side is at least $(k-3)/2$. Hence, since $\boldsymbol{x}_1^{(t_k)}[a_1(k-1)] \geq \boldsymbol{x}_1^{(t_k)}[a_1(k)]$, the gap increases by at least the constant $(k-3)/2$, *i.e.*,

$$\Delta\mathrm{Gap}^{(t_k)}[a_2(k+1)] \; \geq \; \frac{(k-3)}{2}. \tag{42}$$

**Probability mass transfer from $a_1(k-1)$ to $a_1(k)$.** As in (42), we can show that as long as $\boldsymbol{x}_1^{(t)}[a_1(k-1)] \geq \boldsymbol{x}_1^{(t)}[a_1(k)]$, the gap of $a_2(k+1)$ increases by at least $\frac{k-3}{2}$. This inequality, however, cannot hold for the entire period. Indeed, by (40), the gap of $a_1(k)$ decreases by at least $\frac{1}{2}$ in every round in which $\boldsymbol{x}_2^{(t)}[a_2(k)] \geq 1 - \delta$, and therefore the probability mass on $a_1(k)$ increases over time. As a result, there exists a first round $\hat{t}_k$ at which the cumulative utility of $a_1(k)$ overtakes that of $a_1(k-1)$, *i.e.*,

$$\sum_{t=1}^{\hat{t}_k} \boldsymbol{u}_1^{(t)}[a_1(k)] \; \geq \; \sum_{t=1}^{\hat{t}_k} \boldsymbol{u}_1^{(t)}[a_1(k-1)].$$

After $\hat{t}_k$, an identical argument to (40) implies that the gap $\mathrm{Gap}_1^{(t)}[a_1(k-1)]$ increases by at least $1/2$ per round. Our goal is to show that when $\boldsymbol{x}_1^{(t_{k+1})}[a_1(k)] \geq 1 - \delta$, we still have $\boldsymbol{x}_2^{(t_{k+1})}[a_2(k)] \geq 1 - \delta$. Starting from round $\hat{t}_k$, consider $\tau$ additional rounds until round $\hat{t}_k + \tau$. To track the evolution, we outline the timeline of the gaps for $a_1(k-1), a_1(k)$; see Figure 5.

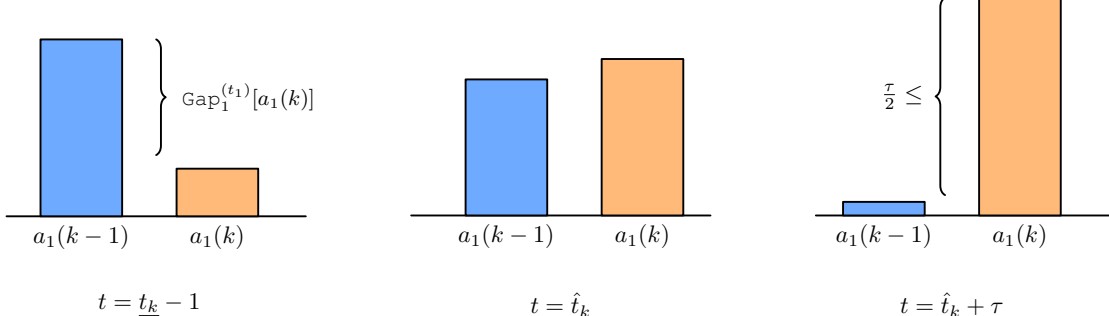

*Figure 5.* Probability mass shift between the two competing actions. The transition from $a_1(k-1)$ to $a_1(k)$ can take a long time when the initial gap is large. Once the cumulative utility of $a_1(k)$ surpasses that of $a_1(k-1)$, the shift in Player 1's behavior triggers the corresponding shift in Player 2's cumulative utility vector. The period-consistency property shows that, throughout the remainder of period $k$, Player 2 assigns high probability only to $a_2(k)$; in particular, no mixing with other strategies occurs.

1. **Time $t_k - 1$.** The cumulative utility of $a_1(k)$ is smaller than that of $a_1(k-1)$.

2. **Time $\hat{t}_k$.** The cumulative utility of $a_1(k)$ first exceeds that of $a_1(k-1)$, so $a_1(k)$ becomes the action of maximal cumulative utility.

3. **Time $\hat{t}_k + \tau$.** The gap of $a_1(k-1)$ has increased to at least $\frac{\tau}{2}$.

Then,

$$\text{Gap}_1^{(\hat{t}_k+\tau)}[a_1(k-1)] \geq \text{Gap}_1^{(\hat{t}_k)}[a_1(k-1)] + \frac{\tau}{2} \geq \frac{\tau}{2}.$$

We next bound the smallest $\tau$ such that, after $\tau$ additional rounds, $a_1(k-1)$ becomes inactive, *i.e.*, $\boldsymbol{x}_1^{(t_2+\tau+1)}[a_1(k-1)] \leq \frac{\delta}{m}$. By Lemma 4.1, it suffices to require

$$\boldsymbol{x}_1^{(\hat{t}_k+\tau+1)}[a_1(k-1)] \leq \frac{R}{\eta^{(\hat{t}_k+\tau)}\,\text{Gap}_1^{(\hat{t}_k+\tau)}[a_1(k-1)]} \leq \frac{R}{\eta^{(\hat{t}_k+\tau)}\,(\tau/2)} \leq \frac{\delta}{m}. \tag{43}$$

Setting $c := \frac{Rm}{\delta}$ and using $\eta^{(t)} = t^{-\alpha}$, (43) is equivalent to

$$\frac{(\hat{t}_k+\tau)^\alpha}{\tau} \leq \frac{1}{2c}. \tag{44}$$

A sufficient condition is obtained by bounding $(\hat{t}_k+\tau)^\alpha \leq 2^\alpha\left(\hat{t}_k^\alpha + \tau^\alpha\right) \leq 2\left(\hat{t}_k^\alpha + \tau^\alpha\right)$, for $\alpha \in [0,1)$. Substituting into (44) concludes the sufficient requirement $(\hat{t}_k^\alpha/\tau) + \tau^{\alpha-1} \leq (1/4c)$. To simplify the algebra, we impose the stronger pair of inequalities $\hat{t}_k^\alpha/\tau \leq 1/(8c)$ and $\tau^{\alpha-1} \leq 1/(8c)$. These are equivalent to

$$\tau \geq 8c\,\hat{t}_k^\alpha \qquad \text{and} \qquad \tau \geq (8c)^{\frac{1}{1-\alpha}},$$

since $\alpha - 1 < 0$ implies $\tau^{1-\alpha} \geq 8c$. Therefore,

$$\tau = \max\left\{(8c)\,\hat{t}_k^\alpha,\ (8c)^{\frac{1}{1-\alpha}}\right\}.$$

Moreover, Lemma C.10 together with (20) implies $\hat{t}_k \geq t_k \geq T_4 \geq \gamma^{(1)}/2 \geq (64c)^{\frac{1}{1-\alpha}}$. Hence, the maximum is attained by the first term. In particular, this shows that it takes at most $\tau$ rounds for $\boldsymbol{x}_1^{(t)}[a_1(k-1)]$ to become inactive, where

$$\tau := (8c)\,\hat{t}_k^\alpha = \left(\frac{8\,Rm}{\delta}\right)\hat{t}_k^\alpha. \tag{45}$$

It remains to show that, even after these $\tau$ rounds, Player 2 still assigns probability at least $1 - \delta$ to action $a_2(k)$. As argued previously for Player 2, every action other than $a_2(k+1)$ accumulates a gap that is linear in $\hat{t}_k + \tau$; hence, by Lemma C.12,

each such action remains assigned probability at most $\delta/m$. Therefore, if $\boldsymbol{x}_2^{(\hat{t}_k+\tau)}[a_2(k)]$ were to drop below $1 - \delta$, then only $a_2(k+1)$ could have increased its probability.

To this end, we lower bound the gap of the competing action $a_2(k+1)$. As noted above, the implication in (42) does not hold after round $\hat{t}_k$. We therefore bound the worst-case one-step change in the gap of $a_2(k+1)$. From (41), for any $t \geq \hat{t}_k$,

$$
\begin{aligned}
\Delta\mathsf{Gap}^{(t)}[a_2(k+1)] &= -\boldsymbol{x}_1^{(t)}[a_1(k)] + \boldsymbol{x}_1^{(t)}[a_1(k-1)](k-1) - \boldsymbol{x}_1^{(t)}[a_1(k+2)](k+2) \\
&\geq -\boldsymbol{x}_1^{(t)}[a_1(k)] - \boldsymbol{x}_1^{(t)}[a_1(k+2)](k+2) \\
&\geq -1 - (\delta/m)(k+2) \geq -2.
\end{aligned}
\tag{46}
$$

By Lemma 4.4, (42) and (46), we obtain

$$
\begin{aligned}
\mathsf{Gap}_2^{(\hat{t}_k+\tau)}[a_2(k+1)] &= \mathsf{Gap}_2^{(\hat{t}_k)}[a_2(k+1)] + \sum_{t=\hat{t}_k+1}^{\hat{t}_k+\tau} \Delta\mathsf{Gap}^{(t)}[a_2(k+1)] \\
&\geq \mathsf{Gap}_2^{(t_k-1)}[a_2(k+1)] + \sum_{t=t_k}^{\hat{t}_k} \Delta\mathsf{Gap}^{(t)}[a_2(k+1)] + \sum_{t=\hat{t}_k+1}^{\hat{t}_k+\tau}(-2) && \text{(From (46))} \\
&\geq \mathsf{Gap}_2^{(t_k-1)}[a_2(k+1)] + \sum_{t=t_k}^{\hat{t}_k} \frac{(k-3)}{2} + \tau(-2) && \text{(From (42))} \\
&\geq \sum_{\ell=4}^{k-1}(\ell-2)T_\ell + \frac{(\hat{t}_k - t_k + 1)}{2} - 2\tau && \text{(From Lemma 4.4)} \\
&\geq \sum_{\ell=4}^{k-1} T_\ell + \frac{(\hat{t}_k - t_k + 1)}{2} - 2\tau \\
&\geq \frac{t_k - 1}{2} + \frac{(\hat{t}_k - t_k + 1)}{2} - 2\tau \\
&= \frac{\hat{t}_k}{2} - (16c)\,\hat{t}_k^\alpha,
\end{aligned}
\tag{47}
$$

where in the last step we used $\tau = (8c)\,\hat{t}_k^\alpha$ from (45).

The correction term $(8c)\,\hat{t}_k^\alpha$ is sublinear in $\hat{t}_k$ for $\alpha \in [0,1)$, so after a particular round the linear term should dominate the second term. It suffices to find $t$ such as the following holds

$$
\frac{t}{2} - (16c)\,t^\alpha \;\geq\; \frac{t}{4},
\tag{48}
$$

which is equivalent to $\frac{t}{4} \geq (16c)\,t^\alpha \Leftrightarrow t \geq (64c)^{\frac{1}{1-\alpha}}$. Therefore, (48) holds whenever $\hat{t}_k \geq (64c)^{\frac{1}{1-\alpha}}$. Moreover, due to Lemma C.10 and (20)

$$
\hat{t}_k \geq \sum_{\ell=3}^{k-1} T_\ell \geq T_4 \geq \frac{\gamma^{(1)}}{2} \geq (64c)^{\frac{1}{1-\alpha}},
\tag{49}
$$

so from (47) we conclude

$$
\mathsf{Gap}_2^{(\hat{t}_k+\tau)}[a_2(k+1)] \;\geq\; \frac{\hat{t}_k}{4}.
\tag{50}
$$

The final step is to verify that $\boldsymbol{x}_2^{(\hat{t}_k+\tau)}[a_2(k+1)] \leq \delta/m$. This ensures that the conditions of Definition 4.2 are satisfied, and hence periods do not ovelap despite the propensity of FTRL to mix actions; in particular, period $k$ can only be followed by period $k+1$. To this end, we invoke Lemma 4.1 together with (50).

$$
\boldsymbol{x}_2^{(\hat{t}_k+\tau)}[a_2(k+1)] \;\leq\; \frac{R}{\eta^{(\hat{t}_k+\tau)}\mathsf{Gap}_2^{(\hat{t}_k+\tau)}[a_2(k+1)]} \;\leq\; \frac{R}{(\hat{t}_k+\tau)^{-\alpha}(\hat{t}_k/4)} = \frac{4R\,(\hat{t}_k+\tau)^\alpha}{\hat{t}_k} \;\leq\; \frac{\delta}{m}
\tag{51}
$$

Hence, it suffices to show that $\frac{(\hat{t}_k + \tau)^\alpha}{\hat{t}_k} \leq \frac{1}{4c}$, where $c = \frac{Rm}{\delta}$. From (44), we already have $\frac{(\hat{t}_k + \tau)^\alpha}{\tau} \leq \frac{1}{2c}$. Therefore,

$$\frac{(\hat{t}_k + \tau)^\alpha}{\hat{t}_k} = \frac{(\hat{t}_k + \tau)^\alpha}{\tau} \frac{\tau}{\hat{t}_k} \leq \frac{1}{2c} \frac{\tau}{\hat{t}_k}.$$

Thus, it remains to ensure $\tau/\hat{t}_k \leq 1/2$, which, in turn, implies (51) and concludes the proof. By (45), $\tau = (8c)\,\hat{t}_k^\alpha$, and hence $\tau/\hat{t}_k = 8c\,\hat{t}_k^{-(1-\alpha)} = 8c/\hat{t}_k^{1-\alpha}$. Therefore, the condition $\tau/\hat{t}_k \leq 1/2$ is equivalent to

$$\frac{\tau}{\hat{t}_k} \leq \frac{1}{2} \iff \frac{8c}{\hat{t}_k^{1-\alpha}} \leq \frac{1}{2} \iff \hat{t}_k^{1-\alpha} \geq 16c \iff \hat{t}_k \geq (16c)^{\frac{1}{1-\alpha}},$$

where the last step uses $1 - \alpha > 0$. Since (49) guarantees $\hat{t}_k \geq (64c)^{\frac{1}{1-\alpha}}$, the above condition is satisfied; hence (51) holds as well, completing the proof. $\qquad\square$

## C.7. Proof of Theorem 1.2

Having established in Property 4.3 that the FTRL iterates traverse the periods sequentially (from $k = 3$ up to $k = 2m - 1$), we can now invoke the preceding results to prove the main theorem of Section 4, namely Theorem 1.2. We begin with a lemma showing that, in every period except the last, the FTRL iterates do not reach an $\epsilon$-Nash equilibrium for any $\epsilon \leq \frac{1}{8m}$.

Throughout, we work with the identical-payoff game $(\mathbf{A}, \mathbf{A})$ defined in Section C.2, and we fix $\delta$ and $\gamma^{(1)}$ as in Property C.3.

**Lemma C.13.** *For every $k \in \{3, \ldots, 2m - 2\}$, throughout period $k$ the FTRL iterates $(\boldsymbol{x}_1^{(t)}, \boldsymbol{x}_2^{(t)})$ do not constitute an $\epsilon$-Nash equilibrium for $\epsilon = \frac{1}{8m}$.*

*Proof.* Fix a round $t$ in period $k$, where $3 \leq k \leq 2m - 2$ and $k$ is even. By Definition 4.2,

$$\boldsymbol{x}_1^{(t)}[a_1(k-1)] + \boldsymbol{x}_1^{(t)}[a_1(k)] \geq 1 - \delta, \qquad \boldsymbol{x}_2^{(t)}[a_2(k)] \geq 1 - \delta.$$

Then,

$$\sum_{a_1 \notin \{a_1(k-1), a_1(k)\}} \boldsymbol{x}_1^{(t)}[a_1] \leq \delta, \qquad \sum_{a_2 \neq a_2(k)} \boldsymbol{x}_2^{(t)}[a_2] \leq \delta.$$

Using the uniform bound $\mathbf{A}[a_1, a_2] \leq 2m - 1$ from Item (ii) in Lemma C.2, we upper bound the realized payoff as

$$\langle \boldsymbol{x}_1^{(t)}, \mathbf{A}\boldsymbol{x}_2^{(t)} \rangle = \sum_{a_1, a_2} \boldsymbol{x}_1^{(t)}[a_1] \boldsymbol{x}_2^{(t)}[a_2] \mathbf{A}[a_1, a_2]$$

$$\leq \delta^2(2m-1) + \boldsymbol{x}_2^{(t)}[a_2(k)] \Big( \boldsymbol{x}_1^{(t)}[a_1(k-1)]\, \mathbf{A}[a_1(k-1), a_2(k)]$$

$$+ \boldsymbol{x}_1^{(t)}[a_1(k)]\, \mathbf{A}[a_1(k), a_2(k)] \Big). \tag{52}$$

Since $k$ is even, Item (iv) in Lemma C.2 implies $a_2(k-1) = a_2(k)$, and hence

$$\mathbf{A}[a_1(k-1), a_2(k)] = \mathbf{A}[a_1(k-1), a_2(k-1)] = k - 1.$$

Substituting this identity and $\mathbf{A}[a_1(k), a_2(k)] = k$ into (52) yields

$$\langle \boldsymbol{x}_1^{(t)}, \mathbf{A}\boldsymbol{x}_2^{(t)} \rangle \leq \delta^2(2m-1) + \boldsymbol{x}_2^{(t)}[a_2(k)] \left( \boldsymbol{x}_1^{(t)}[a_1(k-1)]\,(k-1) + \boldsymbol{x}_1^{(t)}[a_1(k)]\,k \right). \tag{53}$$

Consider Player 1 deviating to the pure action $a_1(k)$. By definition, $\mathbf{A}[a_1(k), a_2(k)] = k$ and all other entries, but the $m$th entry, are nonnegative, hence

$$\left\langle \boldsymbol{e}_{a_1(k)}, \mathbf{A}\boldsymbol{x}_2^{(t)} \right\rangle \geq \boldsymbol{x}_2^{(t)}[a_2(k)]k + \boldsymbol{x}_2^{(t)}[m] \left( -2\gamma^{(1)} \right). \tag{54}$$

The probability assigned to the $m$th action can easily be bounded. The key observation is that the gap of action $m$ grows $\gamma^{(1)}$ times faster than the gap of any action whose gap increases linearly with time. In particular,

$$\texttt{Gap}_2^{(t-1)}[m] \geq \gamma^{(1)}\, \texttt{Gap}_2^{(t-1)}[a_2(2m-1)].$$

By Lemma 4.1, we have

$$\boldsymbol{x}_2^{(t)}[m] \leq \frac{R}{\eta^{(t-1)} \, \mathrm{Gap}_2^{(t-1)}[m]} \leq \frac{R}{\eta^{(t-1)} \gamma^{(1)} \, \mathrm{Gap}_2^{(t-1)}[a_2(2m-1)]} \leq \frac{1}{\gamma^{(1)}} \cdot \frac{\delta}{m}, \tag{55}$$

where the last inequality uses Lemma C.12 applied to $a_2(2m-1)$, whose gap is linear in $t-1$.

Combining this bound with (54) yields

$$\langle \boldsymbol{e}_{a_1(k)}, \mathbf{A}\boldsymbol{x}_2^{(t)} \rangle \geq \boldsymbol{x}_2^{(t)}[a_2(k)] \, k - \frac{2\delta}{m}. \tag{56}$$

Since $\boldsymbol{x}_1^{(t)}[a_1(k-1)] + \boldsymbol{x}_1^{(t)}[a_1(k)] \leq 1$, (53) can be rewritten as

$$\langle \boldsymbol{x}_1^{(t)}, \mathbf{A}\boldsymbol{x}_2^{(t)} \rangle \leq \delta^2(2m-1) + \boldsymbol{x}_2^{(t)}[a_2(k)]\Big( (k-1) + \boldsymbol{x}_1^{(t)}[a_1(k)] \Big).$$

Subtracting it from (56) yields

$$\left\langle \boldsymbol{e}_{a_1(k)}, \mathbf{A}\boldsymbol{x}_2^{(t)} \right\rangle - \left\langle \boldsymbol{x}_1^{(t)}, \mathbf{A}\boldsymbol{x}_2^{(t)} \right\rangle \geq -\delta^2(2m-1) - \frac{2\delta}{m} + \boldsymbol{x}_2^{(t)}[a_2(k)] \left( 1 - \boldsymbol{x}_1^{(t)}[a_1(k)] \right). \tag{57}$$

If $\boldsymbol{x}_1^{(t)}[a_1(k)] \leq 1 - \frac{1}{k+1}$, then $1 - \boldsymbol{x}_1^{(t)}[a_1(k)] \geq \frac{1}{k+1}$ and $\boldsymbol{x}_2^{(t)}[a_2(k)] \geq 1 - \delta$, so

$$\left\langle \boldsymbol{e}_{a_1(k)}, \mathbf{A}\boldsymbol{x}_2^{(t)} \right\rangle - \left\langle \boldsymbol{x}_1^{(t)}, \mathbf{A}\boldsymbol{x}_2^{(t)} \right\rangle \geq -\delta^2(2m-1) - \frac{2\delta}{m} + (1-\delta)\frac{1}{k+1}. \tag{58}$$

Substituting $\delta = \frac{1}{4m}$ and using $k+1 \leq 2m$ yields

$$(1-\delta)\frac{1}{k+1} \geq (1-\delta)\frac{1}{2m}.$$

Hence

$$\begin{aligned}
-\delta^2(2m-1) - \frac{2\delta}{m} + (1-\delta)\frac{1}{k+1} &\geq -\delta^2(2m-1) - \frac{2\delta}{m} + (1-\delta)\frac{1}{2m} \\
&= -\frac{2m-1}{16m^2} - \frac{1}{2m^2} + \left(1 - \frac{1}{4m}\right)\frac{1}{2m} \\
&= -\frac{2m-1}{16m^2} - \frac{1}{2m^2} + \frac{1}{2m} - \frac{1}{8m^2} \\
&= \frac{3}{8m} - \frac{9}{16m^2} = \frac{6m-9}{16m^2} \geq \frac{1}{8m}, \qquad \text{(for } m > 2\text{)}
\end{aligned}$$

$$\left\langle \boldsymbol{e}_{a_1(k)}, \mathbf{A}\boldsymbol{x}_2^{(t)} \right\rangle - \left\langle \boldsymbol{x}_1^{(t)}, \mathbf{A}\boldsymbol{x}_2^{(t)} \right\rangle \geq \frac{1}{8m}. \tag{59}$$

Otherwise, $\boldsymbol{x}_1^{(t)}[a_1(k)] > 1 - \frac{1}{k+1}$, and therefore

$$\boldsymbol{x}_1^{(t)}[a_1(k-1)] \leq 1 - \boldsymbol{x}_1^{(t)}[a_1(k)] < \frac{1}{k+1}.$$

Consider player 2 deviating to $a_2(k+1)$. Since $k$ is even, Item (iii) (in Lemma C.2) implies $a_1(k) = a_1(k+1)$. Hence

$$\mathbf{A}[a_1(k), a_2(k+1)] = \mathbf{A}[a_1(k+1), a_2(k+1)] = k+1.$$

Moreover, all payoff entries appearing in the deviation comparison are nonnegative, except for the $m$th entry, so

$$\left\langle \boldsymbol{x}_1^{(t)}, \mathbf{A}\boldsymbol{e}_{a_2(k+1)} \right\rangle \geq \boldsymbol{x}_1^{(t)}[a_1(k)] \, (k+1) + \boldsymbol{x}_1^{(t)}[m] \left( -2\gamma^{(1)} \right). \tag{60}$$

As before, we can bound $x_1^{(t)}[m] \leq \delta/(\gamma^{(1)}m)$ (analogously to the bound on $x_2^{(t)}[m]$). Using this fact, subtracting (53) from $\langle x_1^{(t)}, \mathbf{A}e_{a_2(k+1)}\rangle$, and using $x_2^{(t)}[a_2(k)] \leq 1$, we obtain

$$\langle x_1^{(t)}, \mathbf{A}e_{a_2(k+1)}\rangle - \langle x_1^{(t)}, \mathbf{A}x_2^{(t)}\rangle \geq x_1^{(t)}[a_1(k)](k+1) - \frac{2\delta}{m} - \delta^2(2m-1)$$
$$- \left(x_1^{(t)}[a_1(k-1)](k-1) + x_1^{(t)}[a_1(k)]k\right)$$
$$= -\delta^2(2m-1) - \frac{2\delta}{m} + x_1^{(t)}[a_1(k)] - (k-1)x_1^{(t)}[a_1(k-1)]. \tag{61}$$

Using $x_1^{(t)}[a_1(k)] > 1 - \frac{1}{k+1}$ and $x_1^{(t)}[a_1(k-1)] \leq \frac{1}{k+1}$ yields $x_1^{(t)}[a_1(k)] - (k-1)x_1^{(t)}[a_1(k-1)] \geq \frac{1}{k+1}$, and hence from (61)

$$\left\langle x_1^{(t)}, \mathbf{A}e_{a_2(k+1)}\right\rangle - \left\langle x_1^{(t)}, \mathbf{A}x_2^{(t)}\right\rangle \geq -\delta^2(2m-1) - \frac{2\delta}{m} + \frac{1}{k+1}$$
$$\geq -\delta^2(2m-1) - \frac{2\delta}{m} + \frac{(1-\delta)}{k+1}. \tag{62}$$

In (58) and (59) we bounded the right-hand side from below by $1/(8m)$. The same bound applies to (62), yielding

$$\left\langle x_1^{(t)}, \mathbf{A}e_{a_2(k+1)}\right\rangle - \left\langle x_1^{(t)}, \mathbf{A}x_2^{(t)}\right\rangle \geq \frac{1}{8m}. \tag{63}$$

Combining (59) and (63), every round $t$ of every even period $k \leq 2m-2$ admits a unilateral deviation that improves the payoff by at least $1/(8m)$. The odd-$k$ case is analogous (with the roles of the players swapped), so no iterate that lies in a period $k \leq 2m-2$ can be an $\epsilon$-Nash equilibrium for any $\epsilon \leq \frac{1}{8m}$. □

We now prove the formal version of Theorem 1.2 stated in Section 1.

**Theorem C.14.** *Consider the identical-payoff game* $(\mathbf{A}, \mathbf{A})$, *and suppose both players run* FTRL *with a permutation-invariant regularizer and learning rate* $\eta^{(t)} = t^{-\alpha}$ *for* $\alpha \in [0, 1)$. *Then, for any* $\epsilon \leq \frac{1}{8m}$, *the* FTRL *iterates do not constitute an* $\epsilon$-*Nash equilibrium for at least* $2^{\Omega(m\log m)}$ *rounds.*

*Proof.* The claim follows by combining Property 4.3, Lemma C.13, and Lemma C.11. By Property 4.3, the FTRL iterates sequentially pass through periods $k = 3, \ldots, 2m-3$. For every round $t$ occurring in these periods, Lemma C.13 guarantees the existence of a unilateral deviation that increases payoff by at least $1/(8m)$. Hence, at every such round the iterate fails to be an $\epsilon$-Nash equilibrium for any $\epsilon \leq 1/(8m)$.

It remains to lower bound the time elapsed by the end of period $2m - 3$. By Lemma C.11, the combined duration of periods $2m - 4$ and $2m - 3$ satisfies

$$T_{2m-4} + T_{2m-3} \geq \frac{\gamma^{(1)}}{4}\left(\frac{m-3}{e}\right)^{m-3}.$$

To express the dominant term in base 2, note that

$$\left(\frac{m-3}{e}\right)^{m-3} = 2^{(m-3)\log_2\left(\frac{m-3}{e}\right)} = 2^{(m-3)(\log_2(m-3)-\log_2 e)}.$$

For $m \geq 10$, we have $m - 3 \geq m/2$ and

$$\log_2(m-3) \geq \log_2(m/2) = \log_2 m - 1.$$

Therefore,

$$(m-3)\left(\log_2(m-3) - \log_2 e\right) \geq \frac{m}{2}\left((\log_2 m - 1) - \log_2 e\right)$$
$$= \frac{m}{2}\left(\log_2 m - (1 + \log_2 e)\right).$$

Let $C := 1 + \log_2 e$. For all $m \geq 2^{2C}(< 10)$ we have $\log_2 m - C \geq \frac{1}{2} \log_2 m$, and hence

$$(m - 3)(\log_2(m - 3) - \log_2 e) \geq \frac{m}{4} \log_2 m.$$

Plugging this back yields

$$\left(\frac{m - 3}{e}\right)^{m-3} \geq 2^{\frac{m}{4} \log_2 m} = 2^{\Omega(m \log m)}.$$

$\square$

# D. Properties of $\mathbf{B}_{m,r}$

In this section, we introduce the class of games introduced by Panageas et al. (2023). Specifically, for $m = 4, 6, \ldots$ and $r \in \mathbb{N}$, we define the matrix $\mathbf{B}_{m,r}$ as follows.

**Definition D.1** (Recursive payoff matrix $\mathbf{B}_{m,r}$)**.** Let $m \geq 2$ be even and $r \in \mathbb{N}$. Define $\mathbf{B}_{m,r} \in \mathbb{R}^{m \times m}$ recursively by

$$\mathbf{B}_{m,r} := \begin{bmatrix} (r+1) & 0 & \cdots & 0 & 0 \\ 0 & & & & (r+4) \\ \vdots & & \mathbf{B}_{m-2,r+4} & & \vdots \\ 0 & & & & 0 \\ (r+2) & 0 & \cdots & 0 & (r+3) \end{bmatrix}, \qquad \text{with base case } \mathbf{B}_{2,r} := \begin{bmatrix} r+1 & 0 \\ r+2 & r+3 \end{bmatrix}. \tag{64}$$

**Lemma D.2** (Support entries of $\mathbf{B}_{m,r}$)**.** *Let $m \geq 2$ and $r \in \mathbb{N}$ be even. Then the set of nonzero entries of $\mathbf{B}_{m,r}$ is exactly $\{r+1, r+2, \ldots, r+(2m-1)\}$, and each value in this set appears exactly once.*

*Proof.* We proceed by induction on $m$. For the base case $m = 2$, the matrix $\mathbf{B}_{2,r}$ has exactly three nonzero entries, namely $\{r+1, \ r+2, \ r+(2 \cdot 2 - 1)\}$, each appearing once. Assume the claim holds for $m - 2$, and consider $\mathbf{B}_{m,r}$ with $m \geq 4$. By construction, the only nonzero entries on the outer frame are

$$r + 1 \text{ at } (1, 1), \qquad r + 2 \text{ at } (m, 1), \qquad r + 3 \text{ at } (m, m), \qquad r + 4 \text{ at } (2, m),$$

and all other outer entries are zero. The inner block is $\mathbf{B}_{m-2,r+4}$, supported on rows and columns in $\{2, \ldots, m-1\}$. By the induction hypothesis, its nonzero entries are exactly

$$\{(r+4) + 1, \ldots, (r+4) + (2(m-2) - 1)\} = \{r+5, \ldots, r + (2m-1)\},$$

each appearing exactly once. Since $\{r+1, r+2, r+3, r+4\}$ is disjoint from $\{r+5, \ldots, r+(2m-1)\}$, all nonzero entries of $\mathbf{B}_{m,r}$ are distinct, and their union is precisely $\{r+1, \ldots, r+(2m-1)\}$. $\square$

**Definition D.3** (Locator functions $a_1, a_2$)**.** Let $m \geq 2$ be even and let $r \in \mathbb{N}$. For each value $s \in \{r+1, \ldots, r+(2m-1)\}$ that appears as an entry of $\mathbf{B}_{m,r}$, define $(a_1(s), a_2(s)) \in [m] \times [m]$ as the unique pair of indices (guaranteed by Lemma D.2) such that

$$\mathbf{B}_{m,r}[a_1(s), a_2(s)] = s.$$

**Proposition D.4** (Structural properties of $\mathbf{B}_{m,r}$)**.** *Let $m \geq 2$ and $r \in \mathbb{N}$ be even. The following properties hold for $\mathbf{B}_{m,r}$:*

*(i)* $\max_{i,j} \mathbf{B}_{m,r}[i, j] = r + (2m - 1).$

*(ii) For even $k \in \{r+1, \ldots, r+(2m-2)\}$, $a_1(k) = a_1(k+1)$.*

*(iii) For odd $k \in \{r+1, \ldots, r+(2m-2)\}$, $a_2(k) = a_2(k+1)$.*

*Proof.* Item (i) is immediate. By Lemma D.2, the nonzero entries of $\mathbf{B}_{m,r}$ are exactly $\{r+1, r+2, \ldots, r+(2m-1)\}$, and hence the maximum is $r + (2m - 1)$.

We prove Items (ii) and (iii) simultaneously by induction on $m$. For $m = 2$, and since $r$ is even, we have $a_2(r+1) = a_2(r+2) = 1$ and $a_1(r+2) = a_1(r+3) = 2$, establishing the claim.

Assume the claim holds for $m - 2$ (with $m \geq 4$), and consider $\mathbf{B}_{m,r}$. By construction,

$$r + 1 \text{ is at } (1, 1), \quad r + 2 \text{ is at } (m, 1), \quad r + 3 \text{ is at } (m, m), \quad r + 4 \text{ is at } (2, m),$$

and the submatrix on rows/cols $\{2, \ldots, m - 1\}$ is $\mathbf{B}_{m-2,r+4}$.

Fix $k \in \{r + 1, \ldots, r + (2m - 2)\}$.

- If $k \in \{r + 1, r + 2, r + 3\}$, the conclusions follow by inspection:

$$a_2(r + 1) = a_2(r + 2) = 1, \quad a_1(r + 2) = a_1(r + 3) = m, \quad a_2(r + 3) = a_2(r + 4) = m.$$

  Thus, for odd $k \in \{r + 1, r + 3\}$ we have $a_2(k) = a_2(k + 1)$, and for even $k = r + 2$ we have $a_1(k) = a_1(k + 1)$.

- If $k = r + 4$, then $r + 4$ is at $(2, m)$ and $r + 5$ is the top-left entry of the inner block, hence at $(2, 2)$. Therefore $a_1(r + 4) = a_1(r + 5) = 2$, which is exactly Item (ii) (since $r + 4$ is even).

- If $k \geq r + 5$, then $k$ and $k + 1$ are inside the inner block $\mathbf{B}_{m-2,r+4}$. Let $(\tilde{a}_1(\cdot), \tilde{a}_2(\cdot))$ denote the location functions within that $(m - 2) \times (m - 2)$ block. Because the inner block is embedded with a $(+1, +1)$ index shift,

$$a_1(t) = \tilde{a}_1(t) + 1, \quad a_2(t) = \tilde{a}_2(t) + 1 \quad \text{for all } \tilde{r} \in \{r + 5, \ldots, r + (2m - 1)\}.$$

  Applying the induction hypothesis to $\mathbf{B}_{m-2,r+4}$ yields $\tilde{a}_1(k) = \tilde{a}_1(k + 1)$ when $k$ is even, and $\tilde{a}_2(k) = \tilde{a}_2(k + 1)$ when $k$ is odd. Shifting by $+1$ gives the desired equalities for $a_1, a_2$ in $\mathbf{B}_{m,r}$.

This completes the induction and proves Items (ii) and (iii). $\qquad \square$

## E. Auxiliary lemmas

**Lemma E.1.** *Let $\mathbf{x}_i \in \Delta(\mathcal{A}_i)$. Suppose there exist $a_i, a_i' \in \mathcal{A}_i$ such that $\mathbf{x}_i[a_i] + \mathbf{x}_i[a_i'] \geq 1 - \delta$ for $\delta = \frac{1}{4m}$. Then*

$$\max\{\mathbf{x}_i[a_i], \mathbf{x}_i[a_i']\} \geq \mathbf{x}_i[\tilde{a}_i] \quad \text{for all } \tilde{a}_i \in \mathcal{A}_i.$$

*In particular, either $\mathbf{x}_i[a_i]$ or $\mathbf{x}_i[a_i']$ is a maximal coordinate of $\mathbf{x}_i$.*

*Proof.* Since $\mathbf{x}_i$ is a probability vector,

$$\sum_{\tilde{a}_i \in \mathcal{A}_i \setminus \{a_i, a_i'\}} \mathbf{x}_i[\tilde{a}_i] = 1 - (\mathbf{x}_i[a_i] + \mathbf{x}_i[a_i']) \leq \delta.$$

Hence, for every $\tilde{a}_i \in \mathcal{A}_i \setminus \{a_i, a_i'\}$,

$$\mathbf{x}_i[\tilde{a}_i] \leq \sum_{\hat{a}_i \in \mathcal{A}_i \setminus \{a_i, a_i'\}} \mathbf{x}_i[\hat{a}_i] \leq \delta. \tag{65}$$

On the other hand,

$$\max\{\mathbf{x}_i[a_i], \mathbf{x}_i[a_i']\} \geq \frac{\mathbf{x}_i[a_i] + \mathbf{x}_i[a_i']}{2} \geq \frac{1 - \delta}{2}. \tag{66}$$

Because $\delta = \frac{1}{4m}$, we have $\frac{1-\delta}{2} \geq \delta$, and therefore from (65) and (66)

$$\max\{\mathbf{x}_i[a_i], \mathbf{x}_i[a_i']\} \geq \mathbf{x}_i[\tilde{a}_i] \quad \text{for all } \tilde{a}_i \in \mathcal{A}_i \setminus \{a_i, a_i'\}.$$

The conclusion is trivial for $\tilde{a}_i \in \{a_i, a_i'\}$, so

$$\max\{\mathbf{x}_i[a_i], \mathbf{x}_i[a_i']\} \geq \mathbf{x}_i[\tilde{a}_i] \quad \text{for all } \tilde{a}_i \in \mathcal{A}_i,$$

which implies that either $\mathbf{x}_i[a_i]$ or $\mathbf{x}_i[a_i']$ is a maximal coordinate of $\mathbf{x}_i$. $\qquad \square$

**Lemma E.2.** *For any player $i$ and any $t \geq 1$, it holds that*

$$\max_{a_i \in \mathcal{A}_i} \sum_{\tau=1}^{t} \boldsymbol{u}_i^{(\tau)}[a_i] - \max_{a_i \in \mathcal{A}_i} \sum_{\tau=1}^{t-1} \boldsymbol{u}_i^{(\tau)}[a_i] \ \geq \ \min\left\{\boldsymbol{u}_i^{(t)}[a_i^{(t-1)}], \ \boldsymbol{u}_i^{(t)}[a_i^{(t)}]\right\},$$

*where*

$$a_i^{(t)} \in \arg\max_{a_i' \in \mathcal{A}_i} \sum_{\tau=1}^{t} \boldsymbol{u}_i^{(\tau)}[a_i'], \quad a_i^{(t-1)} \in \arg\max_{a_i' \in \mathcal{A}_i} \sum_{\tau=1}^{t-1} \boldsymbol{u}_i^{(\tau)}[a_i'].$$

*Proof.* Define

$$M_t := \max_{a_i' \in \mathcal{A}_i} \sum_{\tau=1}^{t} \boldsymbol{u}_i^{(\tau)}[a_i'], \qquad M_{t-1} := \max_{a_i' \in \mathcal{A}_i} \sum_{\tau=1}^{t-1} \boldsymbol{u}_i^{(\tau)}[a_i'].$$

By the definition of $M_t$, for every $a_i' \in \mathcal{A}_i$,

$$M_t \geq \sum_{\tau=1}^{t} \boldsymbol{u}_i^{(\tau)}[a_i'].$$

Applying this inequality with $a_i' = a_i^{(t-1)}$, and using the definition of $a_i^{(t-1)}$, yields

$$\begin{aligned}
M_t - M_{t-1} &\geq \sum_{\tau=1}^{t} \boldsymbol{u}_i^{(\tau)}[a_i^{(t-1)}] - \sum_{\tau=1}^{t-1} \boldsymbol{u}_i^{(\tau)}[a_i^{(t-1)}] \\
&= \boldsymbol{u}_i^{(t)}[a_i^{(t-1)}] \\
&\geq \min\left\{\boldsymbol{u}_i^{(t)}[a_i^{(t-1)}], \boldsymbol{u}_i^{(t)}[a_i^{(t)}]\right\}.
\end{aligned}$$

This proves the claim. $\qquad\square$

**Lemma E.3.** *Let $m \geq 2$ and $\delta = \frac{1}{4m}$. Then, for every integer $k$ with $3 \leq k \leq 2m - 1$, it holds that*

$$(1 - \delta)(k - 1) - \delta(2m - 1) \ \geq \ k - 2.$$

*Proof.* Starting from the left-hand side and subtracting $k - 2$, we obtain

$$(1 - \delta)(k - 1) - \delta(2m - 1) - (k - 2) = 1 - \delta\big((k - 1) + (2m - 1)\big) = 1 - \delta(k + 2m - 2).$$

It therefore suffices to show that $1 - \delta(k + 2m - 2) \geq 0$, equivalently $\delta(k + 2m - 2) \leq 1$. Using $\delta = \frac{1}{4m}$ and $k \leq 2m - 1$, we have

$$\delta(k + 2m - 2) = \frac{k + 2m - 2}{4m} \leq \frac{(2m - 1) + 2m - 2}{4m} = \frac{4m - 3}{4m} < 1.$$

Hence $1 - \delta(k + 2m - 2) > 0$, and thus

$$(1 - \delta)(k - 1) - \delta(2m - 1) \geq k - 2, \quad \text{for } 3 \leq k \leq (2m - 1),$$

as claimed. $\qquad\square$

## F. Proofs from Section 5

We conclude with the proofs from Section 5.

**Lemma 5.2.** *For any $k \geq 2$, $T_k \geq (k - 1)T_{k-1} + T_{k-2} + T_{k-3} - \sum_{\kappa=1}^{k-4} \kappa T_\kappa$. ($T_k$ for $k \leq 0$ is to be interpreted as $0$.)*

*Proof.* We let $\mathcal{T}_k$ be the set of time indices corresponding to $T_k$. Let $i$ be the unique player who has a positive best-response gap during $\mathcal{T}_k$; uniqueness follows from the snake property of $P$ and the construction of the game. We denote by $a_i(k) \in \{0, 1\}$ the action of player $i$ corresponding to payoff $k$, and similarly for $a_i(k + 1)$. $\mathcal{T}_k$ lasts as long as it takes so that $\sum_{\tau=1}^{t} \boldsymbol{u}_i^{(\tau)}[a_i(k + 1)] > \sum_{\tau=1}^{t} \boldsymbol{u}_i^{(\tau)}[a_i(k)]$. This is so because FP picks with probability 1 the action with the highest

cumulative utility; without any loss, we assume for convenience that ties are broken adversarially, so as to maximize the number of steps.

Now, for all times $t$ during the iterations in $\mathcal{T}_{k-1}$, we have $\boldsymbol{u}_i^{(t)}[a_i(k+1)] = \boldsymbol{u}_i^{(t)}[a_i(k)] - (k-1)$. This holds because during those iterations $i$ was obtaining $(k-1)$ by playing $a_i(k)$ while switching to $a_i(k+1)$ would result in $0$ utility since the corresponding action is off the path.

Furthermore, we claim that during all times in $\mathcal{T}_{k-2}$ and $\mathcal{T}_{k-3}$, it holds that $\boldsymbol{u}_i^{(t)}[a_i(k+1)] \leq \boldsymbol{u}_i^{(t)}[a_i(k)] - 1$. Specifically, during $\mathcal{T}_{k-2}$, there exists some player $i' \neq i$ with positive best-response gap, for otherwise the snake property of $P$ would be violated—there would exist an edge between the vertex corresponding to $k-2$ and the vertex corresponding to $k$. This means that during those iterations switching to $a_i(k+1)$ would lead to a utility smaller by at least $1$. During $\mathcal{T}_{k-3}$, it is possible that $i$ has a positive best-response gap, in which case player $i$ eventually transitions from $a_i(k+1) = a_i(k-3)$ (actions are binary and $a_i(k+1) \neq a_i(k)$) to $a_i(k)$, which means that again $a_i(k)$ is the preferred action and satisfies $\boldsymbol{u}_i^{(t)}[a_i(k)] \geq \boldsymbol{u}_i^{(t)}[a_i(k+1)] + 1$. If it is not player $i$ who transitions, playing action $a_i(k+1)$ leads off the path, so the same inequality applies.

As a result, if $t$ is the last round in $\mathcal{T}_k$,

$$\sum_{\tau=1}^{t} \boldsymbol{u}_i^{(\tau)}[a_i(k+1)] - \sum_{\tau=1}^{t} \boldsymbol{u}_i^{(\tau)}[a_i(k)] \leq T_k - (k-1)T_{k-1} - T_{k-2} - T_{k-3} + \sum_{\kappa=1}^{k-4} \kappa T_\kappa.$$

Here we used the fact that $\boldsymbol{u}_i^{(t)}[a_i(k+1)] = \boldsymbol{u}_i^{(t)}[a_i(k)] + 1$ for all iterations $t$ in $\mathcal{T}_k$, and during $\mathcal{T}_\kappa$ we have $\boldsymbol{u}_i^{(t)}[a_i(k+1)] \leq \boldsymbol{u}_i^{(t)}[a_i(k)] + \kappa$. Since $t$ is the last round in $T_k$, it follows that $\sum_{\tau=1}^{t} \boldsymbol{u}_i^{(\tau)}[a_i(k+1)] - \sum_{\tau=1}^{t} \boldsymbol{u}_i^{(\tau)}[a_i(k)] > 0$. This implies

$$T_k \geq (k-1)T_{k-1} + T_{k-2} + T_{k-3} - (k-4)T_{k-4} - \cdots - 1T_1,$$

as claimed. $\qquad\square$

**Lemma 5.3.** *Any sequence satisfying the recurrence relation of Lemma 5.2 grows at least as $(k-1)!$ for $k \geq 1$.*

*Proof.* We define, for $k \geq 2$,
$$S_k := T_k - (k-1)T_{k-1}.$$

It suffices to prove that $S_k \geq 0$ for all $k \geq 2$, as it would imply $T_k \geq (k-1)T_{k-1}$, which in turn yields $T_k \geq (k-1)!$ since $T_3 = 2!$ Using the recurrence relation for $T_k$,

$$S_k = T_k - (k-1)T_{k-1}$$

$$\geq T_{k-2} + T_{k-3} - \sum_{j=1}^{k-4} jT_j. \tag{67}$$

Let $R_k$ denote the right-hand side of (67). We proceed by strong induction. We assume $R_\kappa \geq 0$ for all $2 \leq \kappa < k$, which in turn implies $T_\kappa \geq (\kappa-1)T_{\kappa-1}$ for $2 \leq \kappa < k$. We will show that $R_k \geq 0$. The basis of the induction $k \in \{2, 3, 4\}$ follows trivially, so we take $k \geq 5$.

We consider the difference

$$R_k - R_{k-1} = \left( T_{k-2} + T_{k-3} - \sum_{j=1}^{k-4} jT_j \right) - \left( T_{k-3} + T_{k-4} - \sum_{j=1}^{k-5} jT_j \right)$$

$$= T_{k-2} - T_{k-4} - (k-4)T_{k-4}$$

$$= T_{k-2} - (k-3)T_{k-4}.$$

By the inductive hypothesis,
$$T_{k-2} \geq (k-3)T_{k-3} \geq (k-3)(k-4)T_{k-4}.$$

So, $R_k - R_{k-1} \geq (k-3)(k-5)T_{k-4} \geq 0$. Given that $R_{k-1} \geq 0$, it follows that $R_k \geq 0$, and the claim follows. $\qquad\square$

