# OpenReview forum: "(Doubly) Exponential Lower Bounds for Follow the Regularized Leader in Potential Games"
_ICML.cc/2026/Conference — ICML 2026 spotlight_

### Official Review · Reviewer_4DwP · 2026-03-10

**Soundness:** 3
**Presentation:** 3
**Significance:** 4
**Originality:** 3
**Overall Recommendation:** 5
**Confidence:** 4

**Summary:**

This paper makes theoretical contributions to understanding the convergence rate of Follow-the-Regularized-Leader (FTRL) dynamics in potential games. It establishes that, for any permutation-invariant regularizer, FTRL (and consequently, Multiplicative Weights Update) can take exponential time (in the number of actions m) to reach an approximate Nash equilibrium in two-player potential games, even under a vanishing learning rate schedule. This result is complemented by a doubly exponential lower bound for Fictitious Play (FP), an extreme case of FTRL, in multi-player games. On the positive side, the authors prove an exponential upper bound for a broad class of alternating, lazy no-regret dynamics converging to equilibrium. Collectively, these results characterize the convergence landscape, providing important understanding of several popular learning dynamics in the potential game setting.

**Compliance With Llm Reviewing Policy:**

Affirmed.

**Final Justification:**

The authors clarify my concern in the review. I am positive about the paper’s acceptance.

**Key Questions For Authors:**

1. In the results of the paper, the epsilon for epsilon-Nash equilibrium is relatively small in theorem 1.3. If we only require a constant approximation or a weaker approximation result (like 1/poly(n)) compared to existing findings, how would this affect the conclusions?

2.	In the paper Anagnostides et al. (2025), an exponential lower bound for two-player potential games is also mentioned. What is the relationship between their result and the results in current submission?

3.	The comparison with the paper by Daskalakis & Pan (2014) does not seem entirely appropriate. The submission emphasizes that the result of Daskalakis & Pan (2014) is an exponential lower bound, while the result in this submission is a double-exponential lower bound. However, the result of Daskalakis & Pan (2014) is based on two-player case, whereas the submission addresses multiple players. Could the relationship between the two be further compared?

**Limitations:**

yes

**Strengths And Weaknesses:**

Strengths

1.	The submission addresses a fundamental question regarding the convergence time complexity of many important algorithms in game theory and online learning—namely, FTRL, MWU and FP—within the classic setting of potential games. The established exponential lower bound for FTRL and the doubly exponential lower bound for FP are interesting results. They fundamentally challenge the prevailing perception of these algorithms as being "efficient" optimizers in potential games, revealing a potential inherent computational barrier when they are viewed as first-order methods. This constitutes an important theoretical contribution.

2.	For no-regret dynamics in potential games, the submission provides a constructive upper bound that, in a meaningful sense, matches the established lower bounds. It demonstrates that a broad class of no-regret dynamics can indeed achieve the convergence rate implied by the lower bound in Theorem 1.2.

Weaknesses

1.	The results presented are primarily negative, establishing strong lower bounds without offering clear pathways or suggestions on how these fundamental barriers might be circumvented in practical settings or under different algorithmic assumptions.

2.	The discussion situating this work within the context of related literature is somewhat insufficient. A more thorough comparison and contrast with key prior results would better clarify the exact advance and the broader implications of the findings. Specific questions regarding these relationships are detailed in the "Questions" section.

---

> ### Author Rebuttal · Authors · 2026-03-29
>
> We thank the reviewer for their service and valuable feedback.
>
> **Reviewer**. *“The results presented are primarily negative, establishing strong lower bounds without offering clear pathways or suggestions on how these fundamental barriers might be circumvented in practical settings or under different algorithmic assumptions.”*
>
> Our lower bounds suggest concrete pathways through different algorithmic assumptions. Specifically, it is known that mirror descent (MD) with a smooth regularizer (such as gradient descent) does converge in polynomial time. Our results show that MD with a non-smooth regularizer or any FTRL algorithm can take exponentially longer to converge. Thus, they have concrete practical implications concerning algorithm selection, justifying the use of algorithms such as gradient descent. We will highlight this takeaway more in the revision.
>
> **Reviewer**. *“In the results of the paper, the epsilon for epsilon-Nash equilibrium is relatively small in theorem 1.3. If we only require a constant approximation or a weaker approximation result (like 1/poly(n)) compared to existing findings, how would this affect the conclusions?”*
>
> When $\epsilon$ is a constant, our construction can be adapted to show a lower bound of $exp(1/\epsilon)$. So, if $\epsilon$ is, say $0.01$, the number of iterations would still be astronomically high. We will include this point in the revision.
>
> **Reviewer**. *“In the paper Anagnostides et al. (2025), an exponential lower bound for two-player potential games is also mentioned. What is the relationship between their result and the results in current submission?”*
>
> As we point out in the related work section, that prior paper considers the regret matching (RM) algorithm, while we consider the whole class of FTRL algorithms, which is fundamentally different from RM. RM is *not* an instance of FTRL, and unlike FTRL algorithms, RM is parameter free. A connecting thread between the two works is that both RM and FTRL satisfy the no-regret property. We will clarify this in the revision.
>
> **Reviewer**. *“The comparison with the paper by Daskalakis & Pan (2014) does not seem entirely appropriate. The submission emphasizes that the result of Daskalakis & Pan (2014) is an exponential lower bound, while the result in this submission is a double-exponential lower bound. However, the result of Daskalakis & Pan (2014) is based on two-player case, whereas the submission addresses multiple players. Could the relationship between the two be further compared?”*
>
> The lower bound of Daskalakis & Pan applies to two-player zero-sum games, while our paper considers the class of potential games. Zero-sum games are fundamentally different from potential games, so our results and technical approach altogether are very different. We are happy to make this more clear if the reviewer thinks that our discussion in Lines 127-135 can be misinterpreted.

---

> > ### Author Rebuttal · Reviewer_4DwP · 2026-04-01
> >
> > Thank you for the clarification on the questions.

---

### Official Review · Reviewer_8HB3 · 2026-03-12

**Soundness:** 4
**Presentation:** 4
**Significance:** 4
**Originality:** 3
**Overall Recommendation:** 4
**Confidence:** 4

**Summary:**

This paper investigates the convergence rate of FTRL dynamics in potential games. It shows that FTRL with a decaying learning rate may require an exponential number of rounds to reach an approximate Nash equilibrium, even in two-player identical-interest games. In addition, the paper proves a doubly exponential lower bound for fictitious play in multi-player potential games. These lower bounds are obtained through novel constructions: a recursively defined payoff matrix that yields the exponential bound, and an embedding of a graph-theoretic snake-in-the-box path that produces the doubly exponential bound. The results substantially improve upon previous polynomial lower bounds and demonstrate that FTRL can be dramatically slower than mirror descent in potential games.

**Compliance With Llm Reviewing Policy:**

Affirmed.

**Final Justification:**

I appreciate the authors' response. It addresses my remaining concerns; I will maintain my score and recommend acceptance.

**Key Questions For Authors:**

1. The paper shows that FTRL can be exponentially slow, whereas MD is known to converge in polynomial time. Since FTRL and MD are equivalent under Legendre regularizers, it would be helpful to clarify how this apparent contradiction is resolved. Does your construction rely on the regularizer not being Legendre, or on the particular choice of learning-rate schedule? Please clarify the relationship between your results and the known equivalence between FTRL and MD.
2. In Theorem 3.4, you provide an exponential upper bound. Can your lower-bound construction be adapted to show that this exponential dependence on (1/\epsilon) is essentially tight for FTRL? If so, it would be helpful to state this explicitly.

**Limitations:**

yes

**Strengths And Weaknesses:**

Strengths：

1. The paper provides a thorough and detailed analysis, supported by numerous lemmas and careful inductive arguments to track the evolution of payoff gaps and strategy probabilities. The proofs appear to be correct and complete, and the extensive appendices cover the technical details in full.
2. The construction method used to obtain the lower limit value is very complex and not easy to understand. The recursive matrix used to calculate the upper limit value of the exponent borrows from previous research on virtual games, but has been carefully modified to adapt to the mixed behavior of FTRL.

Weaknesses:

1. Lower bounds are established for carefully constructed potential games. While this suffices to derive worst-case guarantees, it does not address whether such slow convergence phenomena arise in more natural or broadly studied classes of potential games (e.g., congestion games). The constructions appear somewhat artificial, relying on extremely large payoff gaps and finely tuned parameters.
2. Moreover, the paper does not provide upper bounds on the convergence rate of FTRL in potential games, except for a general exponential upper bound for a lazy variant.

---

> ### Author Rebuttal · Authors · 2026-03-29
>
> We thank the reviewer for their service and valuable feedback.
>
> **Reviewer**. *“The constructions appear somewhat artificial, relying on extremely large payoff gaps and finely tuned parameters.”*
>
> First, as we point out in Remark C.5, our FTRL lower bound applies even when the payoffs are in [-1, 1], so one does not need large payoff gaps. Moreover, our lower bound is robust in terms of how FTRL is tuned: it applies whether the learning rate is large or whether it vanishes at a very fast rate.
>
> **Reviewer**. *“Moreover, the paper does not provide upper bounds on the convergence rate of FTRL in potential games, except for a general exponential upper bound for a lazy variant.”*
>
> While our upper bound applies to a lazy variant of FTRL (in fact, to any lazy instantiation of no-regret dynamics), a compelling aspect of our results is that our upper bound matches (up to polynomial factors in the exponent) our lower bound, giving an almost complete answer for that class of algorithms.
>
> **Reviewer**. *“The paper shows that FTRL can be exponentially slow, whereas MD is known to converge in polynomial time. Since FTRL and MD are equivalent under Legendre regularizers, it would be helpful to clarify how this apparent contradiction is resolved. Does your construction rely on the regularizer not being Legendre, or on the particular choice of learning-rate schedule? Please clarify the relationship between your results and the known equivalence between FTRL and MD.”*
>
> There is no contradiction because MD is known to converge in polynomial time *only for smooth regularizers*, and a smooth regularizer on a compact domain cannot be Legendre (the Legendre property requires the gradient of the regularizer to blow up at the boundary). In fact, our lower bound explains a limitation of the prior positive results for mirror descent by showing that smoothness is necessary: without smoothness, mirror descent—and, by equivalence, FTRL in the Legendre setting—can take exponential time. To summarize, our lower bound applies to i) any member of FTRL (with symmetric regularizer, which is almost always the case) and ii) MD with a Legendre regularizer. We will include this discussion in the revised version, and thank the reviewer for bring up this point.
>
> **Reviewer**. *“In Theorem 3.4, you provide an exponential upper bound. Can your lower-bound construction be adapted to show that this exponential dependence on (1/\epsilon) is essentially tight for FTRL? If so, it would be helpful to state this explicitly.”*
>
> The exponential dependence is essentially tight for a broad regime of $\epsilon$, namely when $\epsilon$ is larger than inverse polynomial (in the dimension). Whether it is tight in the higher-precision regime, where $\epsilon$ is exponentially small, is not clear from our current approach. In fact, this is exactly the main open question we described in the conclusions, as that would imply a doubly exponential lower bound for FTRL as well (just as we showed for fictitious play). We will include this discussion in the revision.

---

> > ### Author Rebuttal · Reviewer_8HB3 · 2026-04-06
> >
> > I appreciate the authors' response. It addresses my remaining concerns; I will maintain my score and recommend acceptance.

---

### Official Review · Reviewer_67WV · 2026-03-12

**Soundness:** 3
**Presentation:** 3
**Significance:** 3
**Originality:** 3
**Overall Recommendation:** 5
**Confidence:** 1

**Summary:**

This paper studies the finite-time convergence behavior of the Follow the Regularized Leader (FTRL) family --- including Multiplicative Weights Update --- within the framework of potential games. In particular, the authors investigate how quickly these dynamics approach equilibrium in this class of games. The authors present a negative result: FTRL can necessitate an exponential number of iterations to reach an $\epsilon$-Nash equilibrium in two-player settings. The paper also establishes an exponential upper bound for a “lazy” no-regret variant, which matches the exponent of their proposed lower bounds.

**Compliance With Llm Reviewing Policy:**

Affirmed.

**Final Justification:**

I appreciated the authors' rebuttals and I keep my positive score.

**Key Questions For Authors:**

- Does the lower-bound construction rely in an essential way on the potential-game structure, or could it be extended to broader classes of games?
- Could the authors comment on what types of practical scenarios potential games are intended to model?

**Limitations:**

yes

**Strengths And Weaknesses:**

I am not an expert in this specific area, so my assessment of the technical aspects may be limited. Below I list the strengths and weaknesses I identified.

**Strengths**
- The paper tackles an important question by studying how fast FTRL converges in potential games and establishes strong negative results on its convergence speed.
- The technical results appear non-trivial and rely on fairly involved constructions, although I do not have enough expertise in this specific area to fully judge their technical difficulty.
- The paper is clearly written and the results are presented in a rigorous way.

**Weaknesses**

- The main weakness I identify is that the lower bound applies specifically to FTRL-type dynamics. While this is an important family of algorithms, the result does not rule out faster convergence for other learning dynamics in potential games, which limits the scope of the contribution.

---

> ### Author Rebuttal · Authors · 2026-03-29
>
> We thank the reviewer for their service and valuable feedback.
>
> **Reviewer**. *“The main weakness I identify is that the lower bound applies specifically to FTRL-type dynamics. While this is an important family of algorithms, the result does not rule out faster convergence for other learning dynamics in potential games, which limits the scope of the contribution.”*
>
> We clarify that other learning dynamics, such as gradient descent, do converge fast in potential games. Our main result establishes a separation between FTRL dynamics and dynamics such as gradient descent.
>
> **Reviewer**. *“Does the lower-bound construction rely in an essential way on the potential-game structure, or could it be extended to broader classes of games?”*
>
> Since we are establishing a lower bound, it automatically applies to any broader class of games. We assume the reviewer asks whether it applies to more structured classes of games? This is an interesting question. One such class of games would be symmetric potential games. Another such class would be congestion or routing games. Our lower bounds likely extend to those classes of games, and we can incorporate that if the reviewer considers it important.
>
> **Reviewer**. *“Could the authors comment on what types of practical scenarios potential games are intended to model?”*
>
> Potential games are one of the most well-studied classes of games in the literature going back to the 1970s (Rosenthal, 1973; Monderer and Shapley, 1996). First, they model scenarios such as routing games, be it Internet routing or traffic routing (Roughgarden and Tardos, 2002), and resource allocation problems. Moreover, they model cooperative settings in which a group of agents with aligned interests tries to coordinate without perfect communication (Claus and Boutilier, 1998). As such, potential games are being studied extensively with renewed interest in multi-agent reinforcement learning (e.g., Leonardos et al., 2022; Ding et al., 2022). We will include this discussion in the revised version to strengthen the motivation of our paper.
>
> References:
>
> Monderer, D. and Shapley, L. S. (1996). Potential games. Games and Economic Behavior, 14(1), 124-143.
>
> Rosenthal, R. W. (1973). A class of games possessing pure-strategy Nash equilibria. International Journal of Game Theory, 2(1), 65-67.
>
> Roughgarden, T. and Tardos, É. (2002). How bad is selfish routing? Journal of the ACM, 49(2), 236-259
>
> Claus, C. and Boutilier, C. (1998). The dynamics of reinforcement learning in cooperative multiagent systems. AAAI.
>
> Leonardos, S., Overman, H., Panageas, I., and Piliouras, G. (2022). Global Convergence of Multi-Agent Policy Gradient in Markov Potential Games. ICLR.
>
> Ding, D., Wei, C. Y., Zhang, K., and Jovanovic, M. R. (2022). Independent Policy Gradient for Large-Scale Markov Potential Games. ICLR.

---

> > ### Author Rebuttal · Reviewer_67WV · 2026-04-02
> >
> > I was positive about the paper’s acceptance, and I still am after the authors’ responses.

---

### Official Review · Reviewer_Luh3 · 2026-03-13

**Soundness:** 4
**Presentation:** 4
**Significance:** 3
**Originality:** 4
**Overall Recommendation:** 5
**Confidence:** 4

**Summary:**

This paper studies the convergence of follow-the-regularized-leader (FTRL) and Fictitious Play (FP) algorithms in potential games. For FP, there is an exponential lower bound. This paper makes the following contributions: (1) they show that FTRL algorithms can also take exponential time to converge to an approximate Nash equilibrium in a two-player potential game, for commonly used regularizers and decreasing learning rate; (2) they show that no-regret learning dynamics executed in an $\varepsilon$-lazy, alternating fasion (only one play updates in one iteration, and only updates when the utility improvement is at least $\varepsilon$), has an exponential upper bound.

**Compliance With Llm Reviewing Policy:**

Affirmed.

**Final Justification:**

I thank the authors for answering my questions. I will keep my positive rating and think it is worth acceptance.

**Key Questions For Authors:**

N/A

**Limitations:**

yes.

**Strengths And Weaknesses:**

Strengths:
1. This paper is technically sound and clearly written. The authors made a good effort to present high-level ideas and intuitions, and to provide illustrative figures that help convey the proof ideas.
2. I think the convergence of FTRL family algorithms in potential games is an important question. Although previous works have established an exponential lower bound for the FP algorithm, the FTRL algorithms are no-regret and more popular in practice. This paper not only improves the lower bound for FP, but also shows an exponential lower bound for FTRL. I think these results are significant for learning in games, where lower-bound results are sparser than upper-bound results.
3. The hardness for FTRL is inspired by the one used for FP in Panageas et al. (2023). However, FP plays only pure strategies, whereas FTRL plays mixed strategies due to regularization. This paper makes an original technical contribution by extending the analysis to the more challenging FTRL cases. The improved double-exponential lower bound for FP uses a connection to a graph-theoretic problem known as "snake in the box," which I found quite interesting.

Weaknesses:
Some minor issues:
1. What is the regret of each player when they follow the lazy version of no-regret learning dynamics? It seems to me that this is not a very natural online learning dynamic.
2. The current analysis is restricted to FTRL, but optimistic methods such as OFTRL have better regret/convergence guarantees. Whether the analysis holds for OFTRL is currently missing and worth discussing. It would be very nice to have this strengthened result.

---

> ### Author Rebuttal · Authors · 2026-03-29
>
> We thank the reviewer for their service and valuable feedback.
>
> **Reviewer**. *“What is the regret of each player when they follow the lazy version of no-regret learning dynamics? It seems to me that this is not a very natural online learning dynamic.”*
>
> The (average) regret of an $\epsilon$-lazy learner can be larger by at most $\epsilon$ relative to the non-lazy learner. This holds because the lazy learner refrains from updating only when its current strategy is already $\epsilon$-optimal relative to the current utility, so its regret is close to the regret of the non-lazy algorithm.
>
> **Reviewer**. *“The current analysis is restricted to FTRL, but optimistic methods such as OFTRL have better regret/convergence guarantees. Whether the analysis holds for OFTRL is currently missing and worth discussing. It would be very nice to have this strengthened result.”*
>
> This is an interesting question! We believe that our analysis can indeed be extended to optimistic FTRL algorithms as well, although certain steps in the argument would need to be adjusted. We will try to incorporate this result in a future version of the paper.

---

> > ### Author Rebuttal · Reviewer_Luh3 · 2026-04-03
> >
> > I thank the authors for answering my questions. I will keep my positive rating.

---

### Decision · Program_Chairs · 2026-04-30

**Decision:**

Accept (spotlight)

**Comment:**

This paper establishes that FTRL can take exponential time to converge to a Nash equilibrium in two-player potential games for any (permutation-invariant) regularizer and potentially vanishing learning rate. By known equivalences, this translates to an exponential lower bound for certain mirror descent counterparts, most notably multiplicative weights update. In multi-player potential games, the paper also shows that fictitious play can take doubly exponential time to reach a Nash equilibrium. This constitutes an exponentially stronger lower bound for the foundational learning algorithm in games.

All the Reviewers agree that the paper should be accepted, and, thus, I recommend acceptance of the paper.